# FDNet: An end-to-end fusion decomposition network for infrared and visible images

Jing Di[1], Li Ren[1]*, Jizhao Liu[2], Wenqing Guo[1], Huaikun Zhange[2], Qidong Liu[3], Jing Lian[1]

1 School of Electronic and Information Engineering, Lanzhou Jiaotong University, Lanzhou, Gansu, China, 2 School of Information Science and Engineering, Lanzhou University, Lanzhou, Gansu, China, 3 School of Information Engineering, Zhengzhou University, Zhengzhou, HeNan, China

⊙ These authors contributed equally to this work.
* 1427594911@qq.com

**Data Availability Statement:** All datasets are available in the TNO public database (https://figshare.com/articles/dataset/TNO_Image_Fusion_Dataset/1008029).

## Abstract

Infrared and visible image fusion can generate a fusion image with clear texture and prominent goals under extreme conditions. This capability is important for all-day climate detection and other tasks. However, most existing fusion methods for extracting features from infrared and visible images are based on convolutional neural networks (CNNs). These methods often fail to make full use of the salient objects and texture features in the raw image, leading to problems such as insufficient texture details and low contrast in the fused images. To this end, we propose an unsupervised end-to-end Fusion Decomposition Network (FDNet) for infrared and visible image fusion. Firstly, we construct a fusion network that extracts gradient and intensity information from raw images, using multi-scale layers, depthwise separable convolution, and improved convolution block attention module (I-CBAM). Secondly, as the FDNet network is based on the gradient and intensity information of the image for feature extraction, gradient and intensity loss are designed accordingly. Intensity loss adopts the improved Frobenius norm to adjust the weighing values between the fused image and the two raw to select more effective information. The gradient loss introduces an adaptive weight block that determines the optimized objective based on the richness of texture information at the pixel scale, ultimately guiding the fused image to generate more abundant texture information. Finally, we design a single and dual channel convolutional layer decomposition network, which keeps the decomposed image as possible with the input raw image, forcing the fused image to contain richer detail information. Compared with various other representative image fusion methods, our proposed method not only has good subjective vision, but also achieves advanced fusion performance in objective evaluation.

## 1 Introduction

Image fusion methods use suitable feature extraction methods and fusion strategies to generate a single image containing key image information. The above methods adopt more than two raw images, which provide complement and redundant characteristics. In the realm of image

**Funding:** Jing Di received funding from the Science and Technology Plan Foundation of Gansu Province of China (grant numbers 22JR5RA360) Jing Lian received funding from the National Natural Science Foundation of China (grant numbers 62061023) and the Distinguished Young Scholars of G Ansu Province of China (grant number 21JR7RA345).

**Competing interests:** We also declare that we no competing interests exist.

fusion, one of the most important topics is infrared and visible image fusion [1], which can effectively extract the complementary redundant information between each raw image and combine them into high-quality stable and informative images. It has critical image processing applications, such as remote sensing [2, 3], target detection [4, 5], security surveillance [6], medical imaging [7, 8], and military applications [9].

Infrared and visible image fusion methods can be broadly divided into two categories: traditional methods and deep learning-based methods [10–13]. The traditional methods typically accomplish the image fusion goal in the space domain or frequency domain using corresponding mathematical transformations, such as wavelet transform [14], multiscale transform [15, 16], sparse representation [17]. However, in the image fusion stage, all these methods require manually designed complex image fusion rules. Deep learning-based methods extract and combine image features based on strong feature learning capabilities of neural networks, and could be classified into supervised learningbased methods and unsupervised learning-based methods. Liu et al. [18] adopted the Convolutional Neural Network (CNN) for image fusion and made significant progresses comparing with traditional algorithms, but the CNN requires supervised training. For infrared and visible image fusion tasks, it is impossible to generate usable labeled data. In other words, it is impossible to artificially construct fusion images that can be referenced for supervised training. To address this problem, Li Hui et al [19] proposed to use the pre-trained VGG network for fusing infrared and visible images. This algorithm enables the extraction and fusion of multi-level deep features from the source images. Later, ResNet-50 [20]was proposed to extract and fuse depth features from the source images. However, a significant drawback of these models is their reliance on pre-trained CNN models as offline feature extractors. This limitation prevents adaptive extraction and fusion of features from the source images. Subsequently, scholars designed an end-to-end network framework specifically for image fusion. Prabhakar et al [21]. proposed an unsupervised end-to-end convolutional neural network learning framework, which does not require manually setting complex fusion strategies than other image fusion methods. The novel framework has the flexibility and versatility than previously experienced, but its performance evaluation results are not optimal for specific image fusion tasks.

To solve the above problems, we propose a Fusion Decomposition Network called FDNet to achieve infrared and visible image fusion. Fig 1 shows a set of infrared and visible image pairs, and the corresponding fused images generated by deep learning -based and our proposed FDNet. There are two main aspects for the proposed network. On the one hand, Considering the characteristics of infrared and visible images derived from different sensors, we use multi-scale layers, depthwise separable convolution, and improved Convolutional Block Attention Module(I-CBAM) to create a double-branch network framework to extract the gradient and intensity information of related images. Secondly, we design a new loss function, representing the gradient and intensity information at each image. On the other hand, we consider not only the image fusion process, but also the image decomposition process from the fusion result to raw image. According to the above analysis, we design a single and double channel convolutional layer to maintain consistency between the decomposition result and corresponding raw image. Significantly, the image fusion results can contain a lot of detailed information.

The contribution of our work consists of the following five main aspects:

1. We propose a novel deep learning-based method called as FDNet for fusing infrared and visible images. Compared to traditional image fusion methods, our approach successfully complete image fusion task without manual setting activity level measurement and fusion rules. The overall fusion method can simultaneously perform both the fusion and

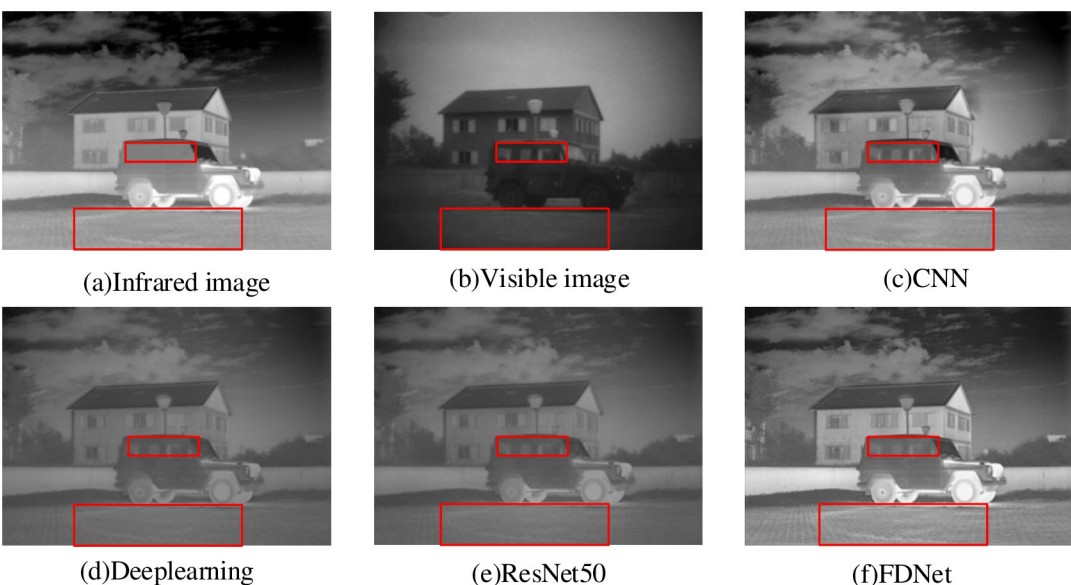

(a)Infrared image    (b)Visible image    (c)CNN

(d)Deeplearning    (e)ResNet50    (f)FDNet

**Fig 1. Schematic illustration of our proposed FDNet through the comparsion with others popular algorithms.** From left to right: the infrared image, the visible image, the fusion results of the CNN [18], the Deeplearning approach [19], the ResNet50 approsch [20], and our proposed FDNet.

decomposition stages. The fusion network designs a double-branch network to complete feature extraction, including multi-scale layers, depthwise separable convolution and I-CBAM. The decomposition network is composed of single and double channel network including convolutional layers, which makes the fusion result contain more scene detail information and improves the network fusion performance.

2. For the shallow feature extraction step, we design multi-scale convolutional network structures to extract image feature information with different receptive field sizes for infrared and visible images. This effectively solves the problem about insufficient feature extraction using a single scale convolution kernel. This not only increases the multiscale convolution structures of the processed image, but also accurately extracts the image features of object regions, and improve the shallow feature extraction ability.

3. For multi-scale deep feature fusion, we design a depthwise separable convolutional structure, which separately considers channel information and spatial information for the image regions. Deep convolution operation and point-by-point convolution operation could guarantee that the size of the feature map is not changed using a deeper network, improve the network expression ability, and build a lightweight network.

4. We propose a novel Frobenius norm loss function, an adaptive gradient loss function, and a structural similarity loss function between the decomposed fused image and the raw image, and generate a desired image fusion result for the novel network.

The remainder of the paper is structured as follows: Section 2 reviews related work. Section 3 presents overall framework, network architecture, and loss functions. Section 4 conducts experimental analyses, algorithm comparisons, and ablation experiments. Section 5 makes conclusion and suggests future work.

## 2 Relevant works

### 2.1 Infrared and visible image fusion

With the emergence of various methods, image fusion techniques have made significant advancements. Currently, the most popular image fusion methods are the deep learning-based methods, which can be further classified into the supervised learning-based methods and unsupervised learning-based methods. It is the most challenging to lack the ground-truth fused images for supervised learning image fusion methods. DeepFuse is the first image fusion method based on unsupervised learning, including encoding step, image fusion step, and decoding step. As the general image fusion framework, the image fusion performance of DeepFuse on specific problems is not good enough. Subsequently, Hui Li et al. [22] proposed DenseFuse, which incorporates the encoder-decoder structure with the Dense-Block and better preserve the original image information. Li et al. [23] proposed NestFuse, a method derived from DenseFuse to retain more detailed features and provide more infrared target information. But finding an effective fusion strategy is difficult for image fusion. To address the issue of arbitrary fusion strategies, Ma et al. [24] proposed the FusionGAN framework based on a genetic algorithm. This image processing framework utilizes a generator to extract and combine meaningful information from the raw images. The purpose of the discriminator in FusionGAN is to enforce the fused image to contain more detailed information in visible image. However, the discriminator network cannot preserve image detail information. Zhang et al. [25] proposed a novel generative adversarial network called GAN-FM, specifically designed to retain more detailed image information. In this network, a full-size jump-connected generator is applied to extract shallow features, and the discriminator uses two Markov discriminators to fully retain the valid information in the infrared and visible images by playing adversarial games with the generator. In addition, a novel intensity masking generative adversarial network (IM GAN) [26] and an unsupervised continual-learning generative adversarial network (UIFGAN) [27] were designed to complement multimodal image information, whereas, it fails to integrate the extracted features efficiently. Xu et al. [12] introduced attention mechanisms to the fusion network for feature extraction, while Liu [28] proposed an Attention-guided and Wavelet-constrained Generative Adversarial Network for infrared and visible image fusion(AWFGAN) model based on Generative Adversarial Nets (GAN), which could better preserve important information of the raw images.

### 2.2 Attention mechanism

Attention mechanism plays an important role for human visual system and brings significant successes in image processing field. The related achievement methods focus selectively on interest regions by reassigning the related weight values of input sequences [29]. Attention mechanism has many image processing applications, including target detection [30], image enhancement [31] and emotion recognition [32], and it can be divided into local-attention, soft-attention and hard-attention [33] according to the achievement methods. Hereinto, Soft-attention mechanism can currently be subdivided into channel attention, spatial attention, and their combined module. Convolutional Block Attention Module (CBAM) is a typical joint module, and its spatial attention model focuses only on important information regions to reduce resource consumption, and the channel attention model allocates channel resources effectively by considering the relationship between feature map channels. Fig 2 shows the block diagram of CBAM.

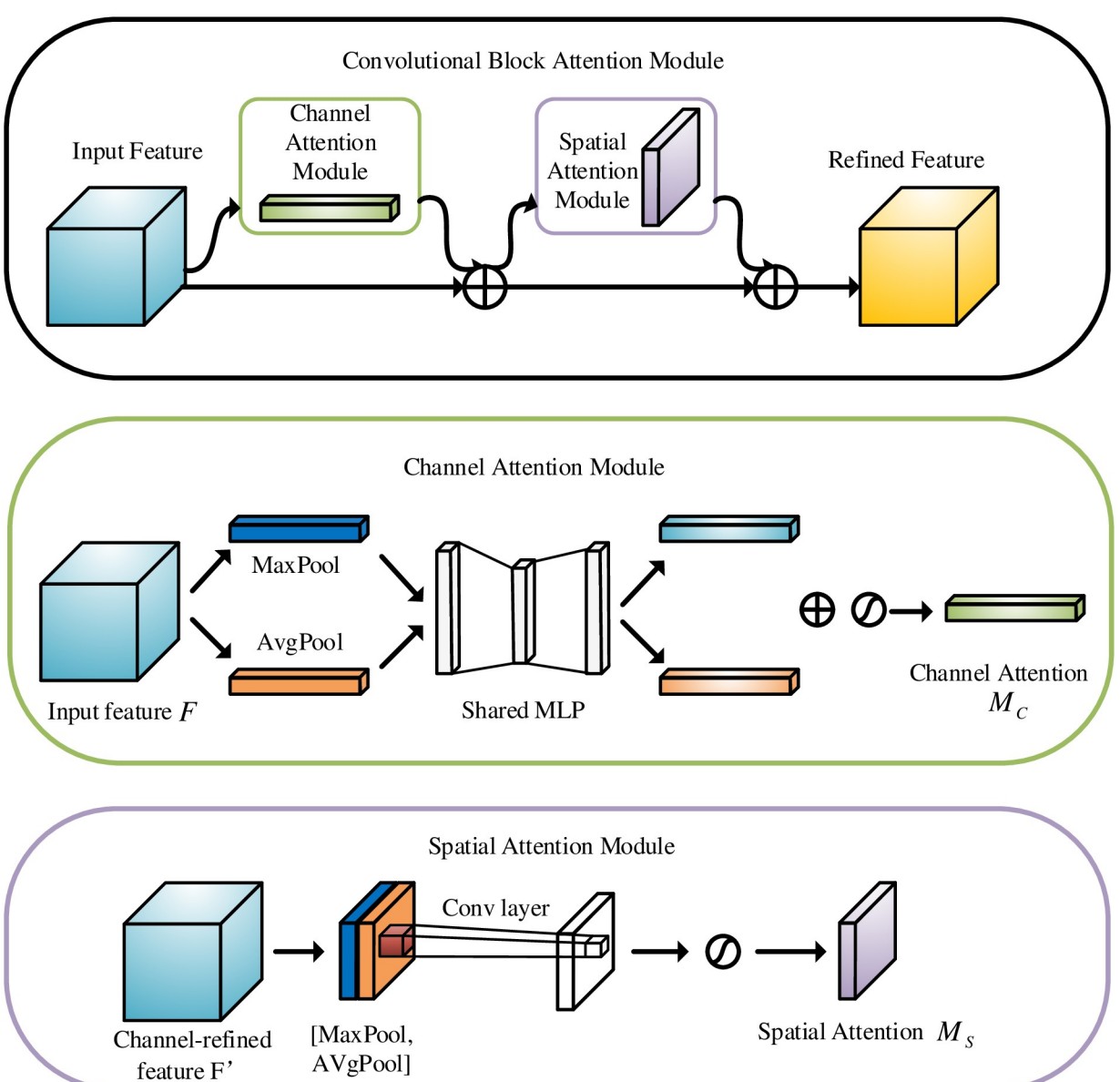

**Fig 2. Block diagram of attention mechanism.**

The related calculating equations about CBAM are written as follows:

$$F' = M_C(F) \otimes F \tag{1}$$

$$F'' = M_S(F') \otimes F' \tag{2}$$

$$
\begin{aligned}
M_C(F) &= \sigma(MLP(AvgPool(F)) + MLP(MaxPool(F))) \\
&= \sigma(W_1(W_0(F_{avg}^C)) + W_1(W_0(F_{max}^C)))
\end{aligned} \tag{3}
$$

$$M_S(F') = \sigma(f^{7\times7}([AvgPool(F'); MaxPool(F')]))$$
$$= \sigma(f^{7\times7}([(F')^S_{avg}; (F')^S_{max}]))$$

$$(4)$$

where $\otimes$ denotes the multiplication operation of corresponding elements. $F$ denotes the input feature map. $F'$ denotes the out result of channel attention mechanism. $F''$ denotes the output result of spatial attention mechanism. $M_C(F)$ denotes the output weights of F based on channel attention. $M_S(F')$ denotes the output weights of $F'$ based on spatial attention. $\sigma$ denotes the sigmoid function. $W_0$ and $W_1$ denote the weight values of the MLP. $F^C_{avg}$ and $F^C_{max}$ denotes the average pooling feature and the maximum pooling feature, respectively. $f^{7\times7}$ denotes a convolution operation with the filter size of $7 \times 7$.

## 2.3 Depthwise separable convolution

The work of Laurent Sifre at Google Brain in 2013 developed depthwise separable convolution (DSC) and was applied to AlexNet to improve recognition accuracy moderately and reduced the size of the proposed model. The first layer of Inception V1 and Inception V2 also used depthwise separable convolution [34, 35]. Within Google, Andrew Howard [36] introduced an efficient mobile models called MobileNets using depthwise separable convolution. Depthwise separable convolution is also a factorization convolution. It has two main achievement steps: depth convolution and pointwise convolution, which are used to filter and combine feature information. This type of factorization not only reduces computational complexity compared to other standard convolutions, but also could acquire better trained models, and is widely used in image classification and image segmentation. Fig 3 illustrates the process of depthwise separable convolution.

## 3 Method

### 3.1 Overall framework

Infrared images have strong anti-interference capability and are not limited by weather conditions. Visible images can provide significant texture and detail information, and have high spatial resolution. In order to enhance the feature extraction capabilities and image fusion performance for visible and infrared images, we proposed FDNet, which is accompanied by its corresponding general framework is shown in Fig 4.

Fig 4, the purple box represents the raw image used for multi-scale feature extraction. The orange box for concat the multi-scale feature extraction map, the blue box represents the depthwise separable convolution operation, the green box for the I-CBAM attention mechanism, the yellow box for concat, the red box for $1 \times 1$ convolution operation and tanh as the activation function, the lavender represents perform $1 \times 1$ convolution and LReLu as the activation function, light green represents perform $3 \times 3$ convolution and LReLu as the activation function, light orange represents perform $3 \times 3$ convolution and tanh as the activation function.

The proposed FDNet fusion network consists of a fusion network and a decomposition network. The fusion network takes into account different properties of raw images from different sensors on the research, so we design a double-branch network to process related data of infrared and visible images, which has large differences from spatial resolution. The purpose of the decomposition network is to contain more abundant scene information for fusion images, so we design a single and double channel convolution layer to obtain a more finely decomposing image. The above proposed network has the same network structure and shared parameters,

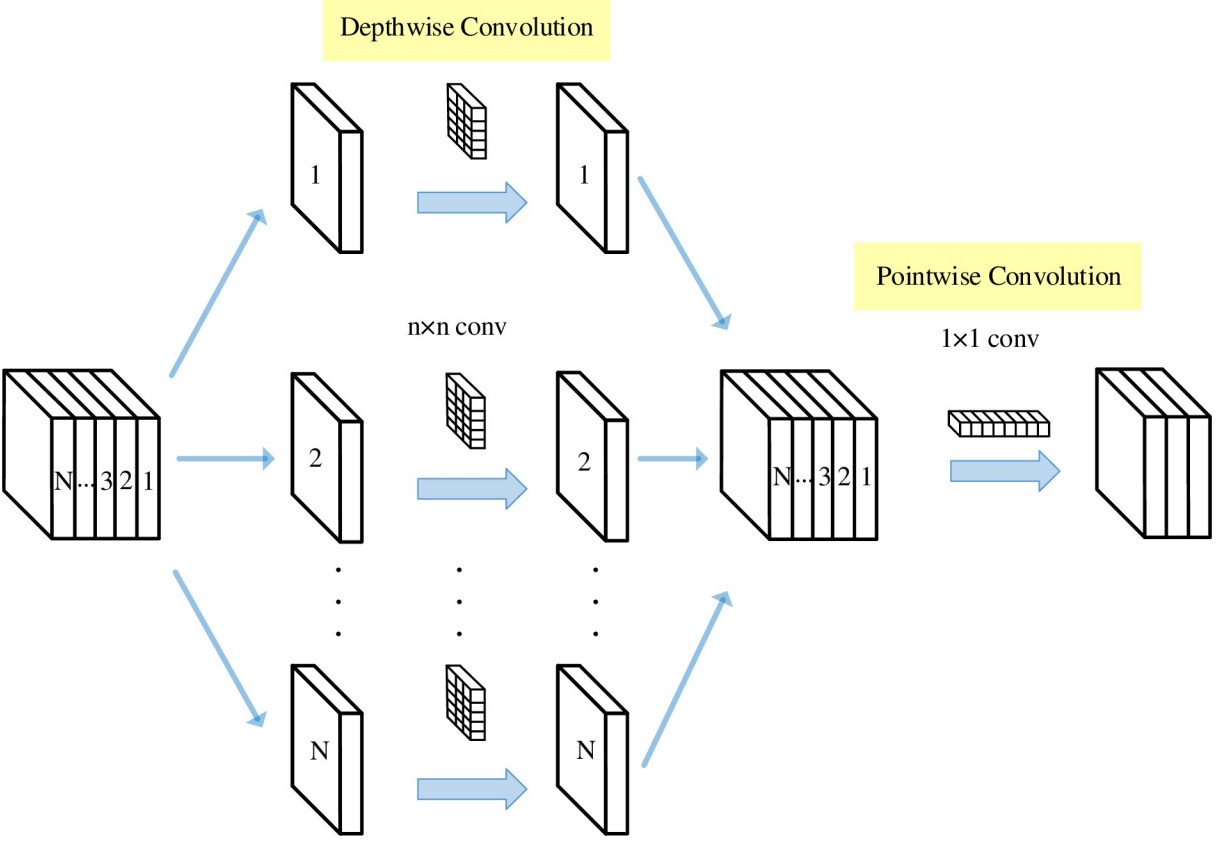

**Fig 3. Depthwise separable convolution process.**

and receives infrared and visible image as the network inputs. The above network structure consists of multi-scale layers, depthwise separable convolution, and I-CBAM. In the training stage, firstly, two modal images with the same size of $120 \times 120$, are entered into the double-branch network. Here its multiscale convolution layer not only extracts the multi-scale features from the raw images, but also reduces the loss of image feature information. The depthwise separable convolution independently conducts spatial convolution step using the multi-scale input features by depthwise convolution operation, and can finds new spatial channels by pointwise convolution operation. Obviously, the related network parameters are reduced and the lightweight network is constructed to achieve deeply feature extraction. Subsequently, I-CBAM focuses on the salient information of infrared and visible images from both channel and spatial aspects, and suppress useless channel information to ensure that all salient features can be utilized during image fusion steps. The extracted features from the double-branch network of infrared and visible images use concat and convolution strategies, shown as a big yellow box in Fig 4. Finally, the decomposition network extracts image feature information by common convolutional layers from the fused images and decomposes them into two branches to generate a new decomposition image consistent with the raw images. In the testing process, the fused images are generated using the trained model data only.

## 3.2 Fusion network

**3.2.1 Shallow feature extraction.** In the deep learning-based methods, feature information is typically extracted using convolutional layers. However, when using a single-scale

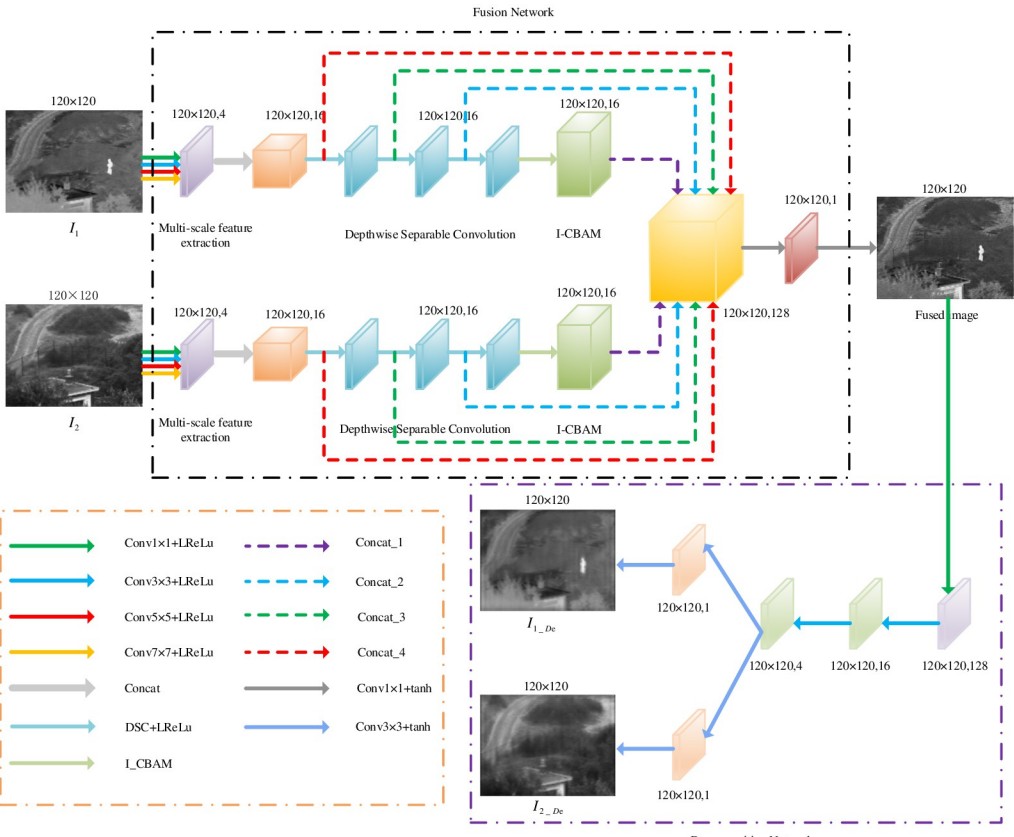

**Fig 4. General framework diagram.**

convolutional kernel, the feature information from other receptive fields may not be fully captured. So, on the research, these convolutional kernels from four different scale sizes with $7 \times 7$, $5 \times 5$, $3 \times 3$, and $1 \times 1$ will be used to extract infrared and visible light image features of different receptive fields, respectively. The multi-scale convolutional layers do not change the size of the raw images, could provide much more image feature information, and extend the ranges of shallow image feature extraction. The structure diagram of multi-scale feature extraction map is shown in Fig 5.

The multi-scale feature extraction equations are calculated as follows:

$$F_j = F_{in} * f_j \tag{5}$$

$$F_{out} = f_{Concat}(F_1, F_3, F_5, F_7) \tag{6}$$

where $F_{in}$ and $F_{out}$ are input feature map and output feature diagram, respectively. $*$ represents the convolution operation. $f_j$ represents the used convolution kernel size (j = 1,3,5,7).

**3.2.2 Deep feature fusion.** The depthwise separable convolution module plays a crucial role in deep feature fusion. Compared to standard convolution operation considering the spatial and channel information in image regions, depthwise separable convolution will consider channel information and spatial regions separately and learn more abundant representation features with less parameters. On the research, we employ the depthwise separable convolution module from the second to fourth layers for deep feature extraction, and select the Leaky Relu

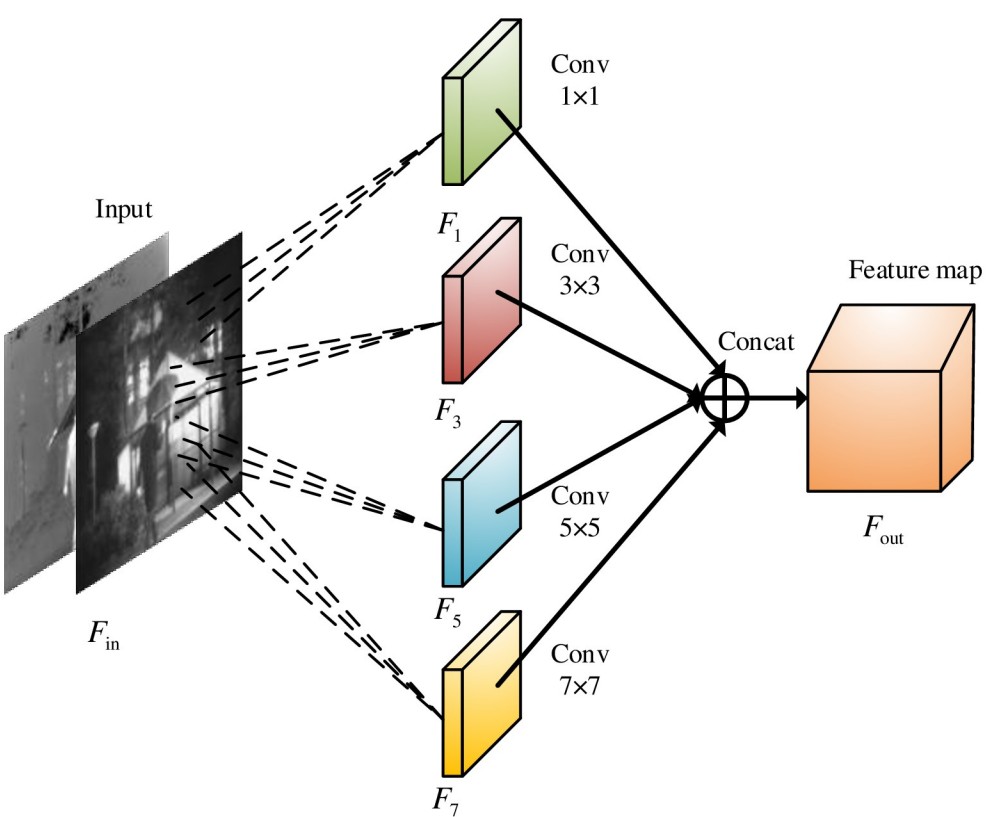

**Fig 5. The structure diagram of multi-scale feature extraction map.**

as the activation function. Firstly, the previous layer in deep feature fusion network mainly adopts 3 × 3 convolution kernels to conduct the spatial convolution operation of each channel and decrease the parameter number. Secondly, the network depth will be deepened by 1 × 1 convolution kernels without changing the size of the feature map, easily realizing cross-channel information interaction and integration, learning deep target information, and improving the network expression capability. The related parameters for the depthwise separable processes are presented in Table 1.

**3.2.3 Improved CBAM.** To enhance the image fusion performance, CBAM is used as the attention module in this study. The receptive field size in the CBAM determines the spatial attention performance. In order to aggregate more extensive spatial context features, a 7 × 7 convolutional kernel in receptive field is used rather than previous 3 × 3 convolution kernel. The number of the module parameters with 7 × 7 convolutional kernel has an obvious increase for the receptive field. Therefore, compared to other same receptive fields, we design a spatial attention module using dilated convolution to complete feature aggregation to reduce the

**Table 1. Depthwise separable process parameter settings.**

| Layer Name | Depthwise Kerbel Size | Pointwise Kerbel Size | Stride | Input Channels | Output Channels | Padding | Activation |
|---|---|---|---|---|---|---|---|
| DSC1 | 3 | 1 | 1 | 16 | 16 | Same | LReLu |
| DSC2 | 3 | 1 | 1 | 16 | 16 | Same | LReLu |
| DSC3 | 3 | 1 | 1 | 16 | 16 | Same | LReLu |

number of module parameters. The specific spatial attention calculation equation is given as follows:

$$
\begin{aligned}
M_S(F) &= \sigma\big(f^{3\times3}_{dilat}([AvgPool(F); MaxPool(F)])\big) \\
&= \sigma\big(f^{3\times3}_{dilat}([(F)^S_{avg}; (F)^S_{max}])\big)
\end{aligned}
\tag{7}
$$

where $F$ denotes the input feature map. $M_S(F)$ denotes the output weights of $F$ based on spatial attention. $\sigma$ denotes the sigmoid function. $f^{3\times3}_{dilat}$ denotes the dilated convolution with a convolution kernel size of 3. The experiments use the dilated convolution with a dilated rate of 2.

The CBAM attention mechanism generally adopts "cascade" connection, but this will bring a large influence that the previous feature mapping determines the later weighing values and learned features from the attention modules. Significantly, the interference caused by the "cascade connection" could bring a worse effect for the attention modules in image fusion tasks. Therefore, we change the original "cascade connection" to "parallel connection", which directly learns the initial input feature map without considering the order of spatial attention and channel attention, and the related mathematical equation is given as:

$$
F'' = M_C(F) \otimes M_S(F) \otimes F
\tag{8}
$$

where $F''$ denotes the final output feature map. $M_C(F)$ denotes the output weights of $F$ based on channel attention.

For the I-CBAM, the spatial attention module and the channel attention module are learned simultaneously. Hereinto, In the channel attention module, the input feature diagram $F(H \times W \times C)$ is subjected to the maximum pooling and average pooling, and obtain two feature diagrams of $1 \times 1 \times C$, and then they are given to a Multi-Layer Perceptron (MLP). The channel feature diagram is generated by element-wise operation and sigmoid activation, known as $M_C$. In the spatial attention module, Firstly, the input feature diagram $F$ obtains two feature diagrams of $H \times W \times 1$ by maximum pooling and average pooling operation. Secondly, we conduct a channel-based concat operation and use the dilated convolution with convolution kernel size 3 to reduce the number of dimensions. Thirdly, through the Sigmoid activation function obtain the final spatial feature diagram $M_S$. Fourthly, the feature map obtained by channel attention and spatial attention is directly weighted with the original input feature map $F$ to obtain the final output feature diagram. The overall block diagram of I-CBAM is shown in Fig 6.

## 3.3 Decomposition networks

The purpose of designing decomposing network is to decompose the fused images and to generate good image fusion results closer to raw images. The framework of the decomposition network is illustrated in Fig 7.

In Fig 7, We extract image features from the fused image using three single-channel convolutional layers, and then generate the decomposition results from two dual-channel convolutional layers. The first convolutional layer utilizes $1 \times 1$ convolutional kernel, and the remaining convolution layers employ a $3 \times 3$ convolutional kernel. The activation function Leakly ReLU is chosen for the common convolutional layer, and the activation function Tanh is adopted for the last double-channel convolutional layer.

## 3.4 Loss functions

FDNet architecture is divided into the fusion and decomposition components. The fusion network combines multiple images into a single fused image through feature extraction.

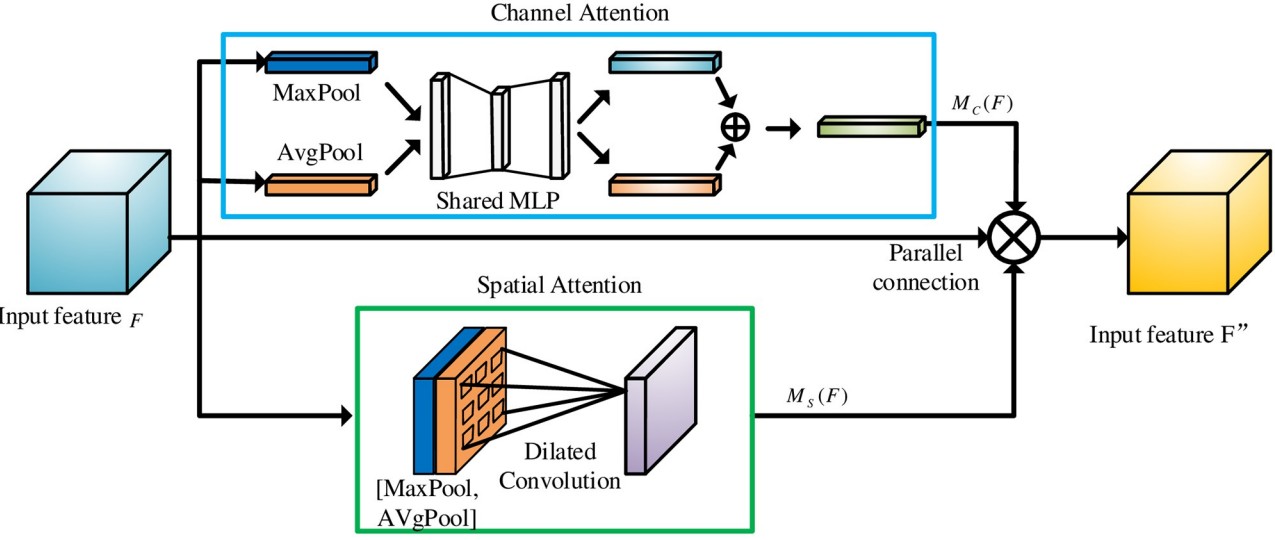

**Fig 6. I-CBAM overall structure diagram.**

Moreover, the decomposition network is to make these fused results contain deeper scene information. The corresponding loss function consists of fusion loss $L_{sf}$ and decomposition loss $L_{dc}$. The mathematical expression is written as:

$$L = L_{sf} + L_{dc} \tag{9}$$

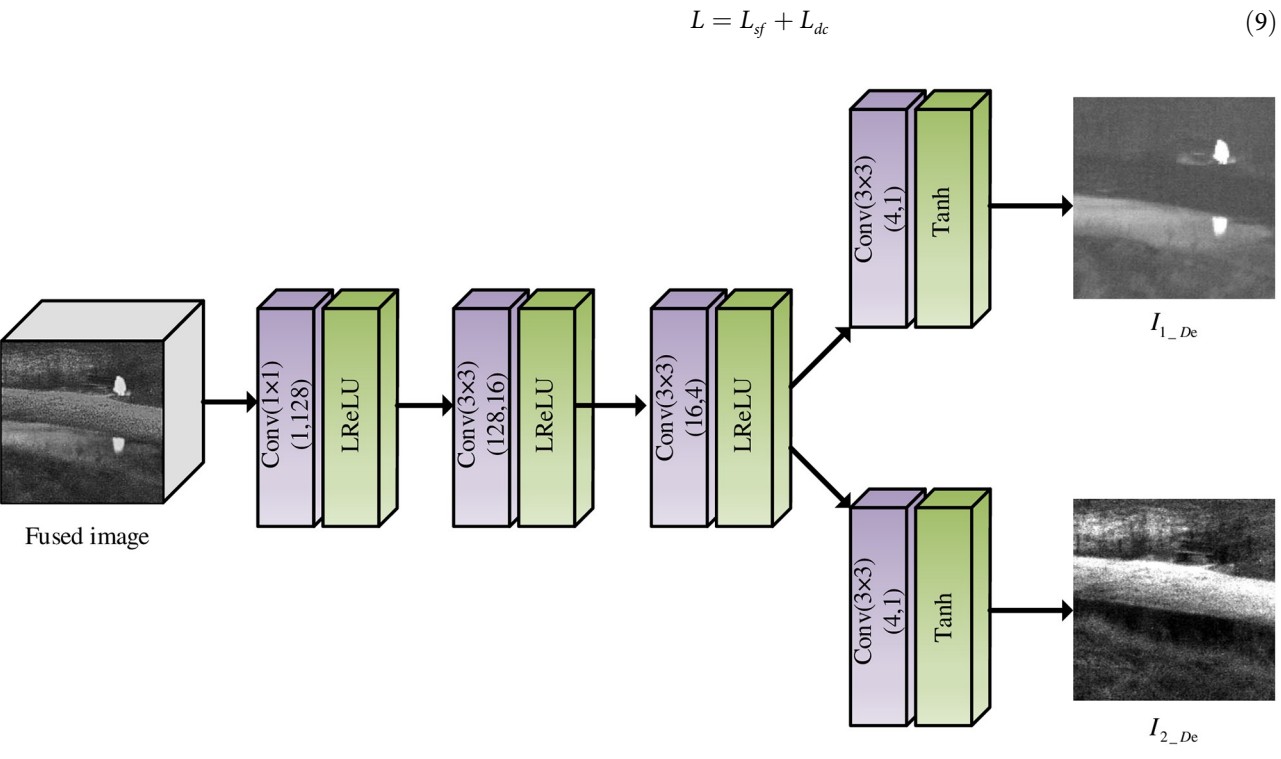

**Fig 7. Decomposition network framework.**

where $L$ represents the total loss function, $L_{sf}$ represents the fusion loss, $L_{dc}$ represents the decomposition loss.

**3.4.1 Fusion loss.** The most basic components of infrared and visible images are image pixels, whose intensities represent the overall pixel luminance distribution. The differences between pixels could form gradient information, which represents the texture details of a raw image. Therefore, the traditional infrared and visible image fusion scheme can be constructed to extract and reconstruct the gradient and intensity information from raw images on the research. Correspondingly, the fusion loss function composes of intensity loss and gradient loss. The corresponding equation is expressed as:

$$L_{sf} = \beta L_{grad} + L_{int} \tag{10}$$

$\beta$ is the key parameter between the intensity term and the gradient term, $L_{grad}$ represents the adaptive gradient loss function, $L_{int}$ represents the intensity function.

An adaptive gradient loss function $L_{grad}$ is designed to add abundant texture features for the fusion images. We also introduce an adaptive weight block to reduce the noise influence by Gaussian low-pass filter on the weighing block. This adaptive weight block evaluates the optimization objectives of the respective pixels in the raw images based on the richness of gradients. The complete process of the adaptive weight block is depicted in Fig 8.

The equations of gradient loss function are expressed as follows:

$$L_{grad} = \frac{1}{HW} \sum_i \sum_j S_{1i,j} \cdot \left( \nabla I_{fused_{i,j}} - \nabla I_{1_{i,j}} \right)^2 + S_{2i,j} \cdot \left( \nabla I_{fused_{i,j}} - \nabla I_{2_{i,j}} \right)^2 \tag{11}$$

$$S_{1i,j} = sign(| \nabla (L(I_{1i,j}))|) - min(| \nabla (L(I_{1i,j}))|, | \nabla (L(I_{2i,j}))|) \tag{12}$$

$$S_{2i,j} = 1 - S_{1i,j} \tag{13}$$

where $I_1$ and $I_2$ are the raw images, $I_{fused}$ is the fused image, $H$ and $W$ denote the height and width of the processing images, respectively. $i$ and $j$ represent pixel coordinates in position ($i$, $j$). $\nabla(\cdot)$ is the Laplacian operator. $L(\cdot)$ and $|\cdot|$ represent the Gaussian low-pass filter function and absolute value function, respectively. $min(\cdot)$ and $sign(\cdot)$ denotes the minimum function and sign function, respectively.

Intensity loss, adopting improved the Frobenius norm, affects the brightness and contrast of the image, and brings the natural and realistic effect for the fused images. The loss function is defined as the square root of the sum of the squares for the matrix elements at each position.

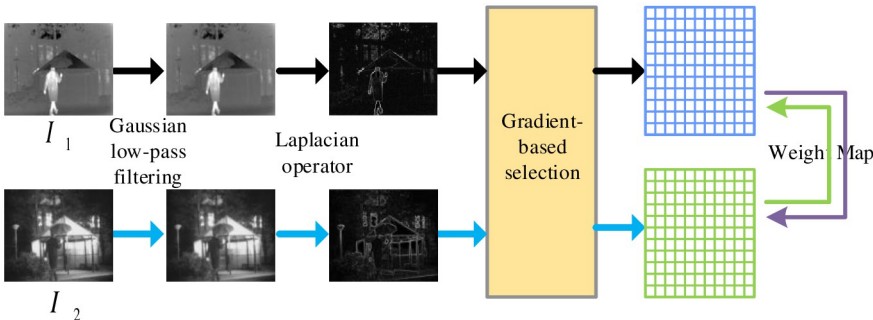

**Fig 8. Schematic diagram of adaptive weight block.**

Its main role is to measure the distance of the matrices between the raw and fused image pixels, and to adjust their weighing values effectively. It is noted that the function could select more effective information in the network training process. The related formula is expressed as follows:

$$L_{int} = \sqrt{\frac{1}{HW}\sum_i\sum_j |I_{fused_{i,j}} - I_{1_{i,j}}|^2 + \alpha |I_{fused_{i,j}} - I_{2_{i,j}}|^2} \tag{14}$$

$\alpha$ is used to adjust the infrared and visible image intensity information.

**3.4.2 Decomposition loss.** Decomposition loss $L_{dc}$ requires that the decomposition result of a raw image after image fusion step is akin to the corresponding raw image. We choose the structural similarity (SSIM) as the loss function, and calculate the SSIM value between the fusion decomposition result and the raw image, in terms of structural distortion, contrast distortion and luminance distortion. The corresponding formulae is written as follows:

$$L_{dc} = \left(1 - \frac{1}{HW}\sum_i\sum_j SSIM\left(I_{1de_{i,j}}, I_{1_{i,j}}\right)\right) + \left(1 - \frac{1}{HW}\sum_i\sum_j SSIM\left(I_{2de_{i,j}}, I_{2_{i,j}}\right)\right) \tag{15}$$

$$SSIM(I_{1de}, I_1) = \frac{2\mu_{I_{1de}}\mu_{I_1} + C_1}{\mu_{I_{1de}}^2 + \mu_{I_1}^2 + C_1} \cdot \frac{2\sigma_{I_{1de}}\sigma_{I_1} + C_2}{\sigma_{I_{1de}}^2 + \sigma_{I_1}^2 + C_1} \cdot \frac{\sigma_{I_{1de},I_1} + C_3}{\sigma_{I_{1de}}^2 \sigma_{I_1}^2 + C_3} \tag{16}$$

$$SSIM(I_{2de}, I_2) = \frac{2\mu_{I_{2de}}\mu_{I_2} + C_1}{\mu_{I_{2de}}^2 + \mu_{I_2}^2 + C_1} \cdot \frac{2\sigma_{I_{2de}}\sigma_{I_2} + C_2}{\sigma_{I_{2de}}^2 + \sigma_{I_2}^2 + C_1} \cdot \frac{\sigma_{I_{2de},I_2} + C_3}{\sigma_{I_{2de}}^2 \sigma_{I_2}^2 + C_3} \tag{17}$$

where $I_{1de}$ and $I_{2de}$ are the decomposition results, $I_1$ and $I_2$ are the raw images. $\mu$ and $\sigma$ are the mean value and standard deviation, respectively. The parameters $C_1$, $C_2$, and $C_3$ are three important constants to avoid the SSIM value to zero during the training process.

## 4 Experimental results and analysis

### 4.1 Datasets and setup details

This paper utilizes the publicly available TNO database for completing infrared and visible image fusion tasks. We add related experimental images by designed cropping and decomposing methods in training steps. The training images with the maximum pixel size of $576 \times 768$ and the minimum pixel size of $360 \times 270$ are selected and cropped to generate 42,484 experimental images with the suitable size of $120 \times 120$. In contrast to the training data, ten pairs of testing images are selected as the testing data, using the original sizes of raw images.

This experiment is conducted on a Windows 10 operating system with an Intel Core i5-1035G1 CPU. Tensorflow and imageio are used to train and test network performance in the Pycharm compiler. The related parameters of the experiments are set to epoch = 15, batch size = 32, learning rate = 1e-4. The strong and well-converging adaptive optimization algorithm Adam are adopted for the optimization algorithms. In Eq 14, the parameter $\alpha$ is 0.5, allowing the network to obtain the main intensity information from the infrared images to maintain high contrast. In addition, after repeated experiments, the ratios of gradient loss, intensity loss, and decomposition loss included in the total loss are set to 80, 1, and 1, respectively.

## 4.2 Ablation experiments

**4.2.1 Module performance test.** To validate the effectiveness of our proposed methods, related ablation experiments are conducted as follows:

1. Image fusion experiments with multi-scale depthwise separable convolution (M-DSC).

2. Image fusion experiments with multi-scale depthwise convolution and attention mechanism (M-DSC+CBAM).

3. Image fusion experiments with multi-scale depthwise convolution and improved attention mechanism (M-DSC+I-CBAM).

An image fusion result of "soldier-behind-smoke-1" was randomly selected from the testing dataset for subjective evaluation. Ten groups of image fusion results were chosen for objective evaluation. The evaluation results of the ablation experiment are shown in Fig 9. It is noted that the overall image contrast is slightly insufficient only using M-DSC, and some image details cannot be captured. The fusion results of M-DSC and CBAM significantly solve these problems, making the person in the forest clearer and improving the overall image contrast to reveal more image details. Furthermore, the image fusion results obtained by combining our proposed M-DSC with I-CBAM effectively retaines ential image information and show significant improvements compared to the previous two comparison methods.

In the ablation experiment, we select the average gradient (AG) and multi-scale structural similarity (MSSSIM) as objective evaluation metrics. The MSSSIM uses different resolutions to evaluate the image fusion quality and reflects the fused image sharpness and texture detail information with AG. Table 2 is the objective evaluation results of adopted metrics for the ablation experiment.

In Table 2, the increase of the number of modules brings obvious improvement for image fusion performance, and the calculation value of multi-scale structure similarity metric is 88.

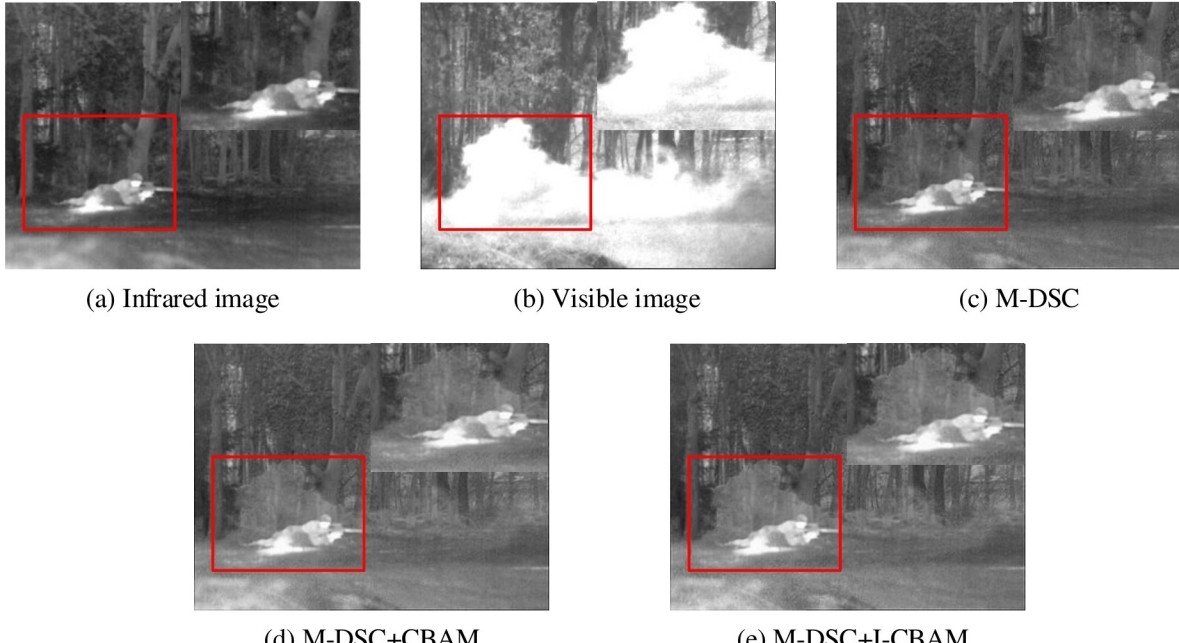

(a) Infrared image          (b) Visible image          (c) M-DSC

(d) M-DSC+CBAM          (e) M-DSC+I-CBAM

**Fig 9. Module ablation experimental results.**

**Table 2. Objective evaluation results of ablation experiments.**

| Moudle | AG | MSSSIM |
|---|---|---|
| M-DSC | 5.7700 | 0.8268 |
| M-DSC+CBAM | 5.8635 | 0.8688 |
| M-DSC+I-CBAM | 6.0663 | 0.8820 |

2%, which demonstrates that the proposed M-DSC and I-CBAM methods can effectively extract image features.

**4.2.2 Decomposition network.** A decomposition network is typically used to decomposes an image fusion result to the corresponding raw image approximately. The decomposition result is determined by the image fusion result and prompt the fused images to acquire more scene details. To demonstrate the effectiveness of the decomposition network for the fused images, we conduct related ablation experiments, as shown in Fig 10.

In Fig 10, the using decomposition network improves the clarity of the trees and soldiers in the image fusion result, and the overall visual effect is better. Additional, two objective evaluation metrics, namely spatial frequency SF and average gradient AG, are chosen to reflect the clarity of the processed images. The experimental data is shown in Table 3.

From Table 3, the fused image clarity in our proposed network becomes higher than missing decomposition network, and the image fusion performance is improved referring to the above experiment data.

**4.2.3 Intensity loss analysis.** Intensity loss plays a key role in the fused image to retain important information, such as contrast. Meanwhile, this method helps to maintain a natural scene style in the fused image. For this reason, we perform the ablation experiments to prove the effectiveness of the proposed method, as shown in Fig 11.

As shown in Fig 11, the lack of intensity loss generates several problems, including low brightness, information loss and stylistic unrealism for the fused images. This indicates that intensity loss is critical for the image fusion results. Due to the significant deviation of the experimental results without intensity loss from the expected outcome,we decided not to conduct quantitative experiments in this case.

**4.2.4 Gradient loss analysis.** Gradient loss forces the more texture details in the fused image, as demonstrated by our ablation experiments. The calculation results of gradient loss ablation experiment are shown in Fig 12. In the given image fusion results with no gradient loss shows texture loss and sharpness reduction, while the use of gradient loss retains original sharpness and acquire more texture details. Additionally, objective evaluation results of gradient loss ablation experiment is given in Table 4.

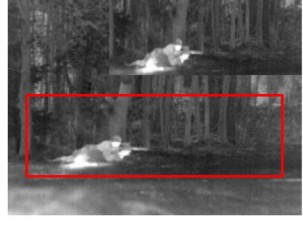 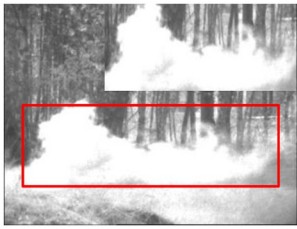 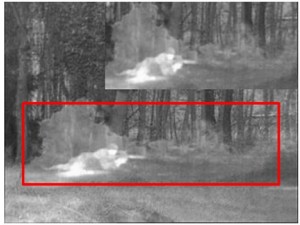 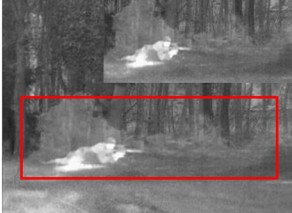

(a)Infrared image (b)Visible image (c)Missing decomposition network (d)Ours

**Fig 10. Decomposition network ablation experiment.**

**Table 3. Objective evaluation results of ablation experiments.**

| Moudle | SF | AG |
|---|---|---|
| Missing decomposition network | 17.5718 | 5.9394 |
| Ours | 17.6834 | 5.9840 |

In Table 4, it can be seen that the inclusion of gradient loss leads to further improvement in the image fusion results. This strongly demonstrates the significance of gradient loss in enhqncing the fusion performance.

### 4.3 Fusion image analysis

Different image evaluation methods are easy to give different evaluation results in most cases. In this study, we adopt subjective and objective evaluation methods to evaluate the image fusion effect of a randomly given image.

**4.3.1 Subjective evaluation.** The proposed fusion method is compared with twelve prevalent methods, including Deeplearning [19], ResNet50 [20], CDL [37], CCFL [38], SMVIF [39], BF [40], MLGCF [41], SDNet [42], SwinFusion [43], PIAFusion [44], FLFusion [45] and U2Fusion [46].

The image fusion result of "Nato camp" is shown in Fig 13. In the BF method, the grass, person and plants around the building are all blurred, and the overall fusion effect is bad. The ResNet50, Deeplearning, SMVIF, U2Fusion and FLFusion methods provide more de tailed information about the persons in the fused image, but the plant is still blur. In the MLGCF,

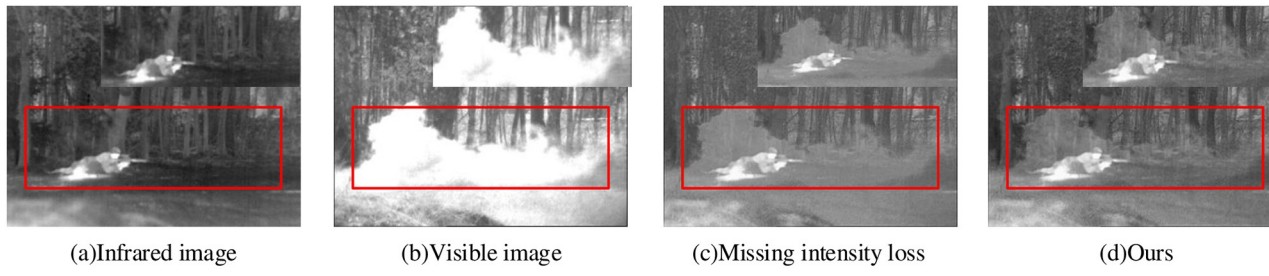

(a)Infrared image     (b)Visible image     (c)Missing intensity loss     (d)Ours

**Fig 11. Intensity loss ablation experiment.**

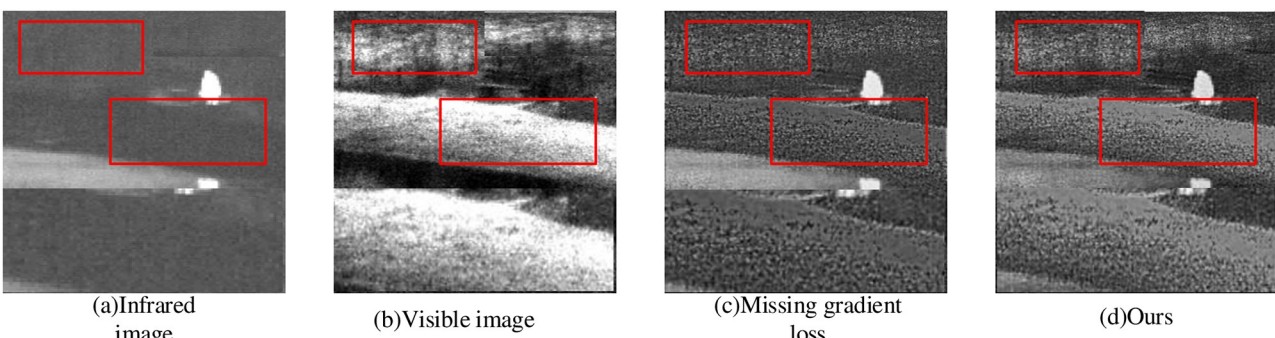

(a)Infrared image     (b)Visible image     (c)Missing gradient loss     (d)Ours

**Fig 12. Gradient loss ablation experiment.**

**Table 4. Objective evaluation results of ablation experiments.**

| Moudle | EN | AG | SD | SF |
|---|---|---|---|---|
| Missing gradient loss | 6.8203 | 13.1001 | 5.8110 | 33.3563 |
| Ours | 6.9858 | 14.1859 | 5.9775 | 36.2779 |

SDNet, SwinFusion and PIAFusion methods, the person and grass are not blurred, but the overall contrast is low, limiting the visibility of additional feature information. The CDL and CCFL methods have better overall contrast, but the target edges are not clear enough, and some detailed information is not clear. Compared to previous popular methods, our proposed method yields clearer texture details, richer scene information and remarkable target objects.

The "helicopter" fusion result is shown in Fig 14. The FLFusion fails tot fully integrate the visible image information, resulting in a fusion result dominated by infrared information. Although the BF method shows some improvement compared to the FLFusion, it still falls short in fully extracting the original visible information, resulting in an image with only contour information. The ResNet50, Deeplearning, SMVIF, MLGC-F and U2Fusion methods maintain the related information of the raw image, the whole image is blurry and the specific texture information is unclear. The SwinFusion and PIAFusion methods have prominent infrared image object regions, but it brings a low overall contrast, which leads to the inability to represent more detailed information. The CDL and CCFL methods have better overall brightness, but the clarity is not high with artifacts around the target. The SDNet method shows significant targets and clear textures, but some missing edge information. In comparison, our proposed method retains the feature information while obtaining the best brightness and largest edge gradient, with clear background texture and good visual effect for the processed image.

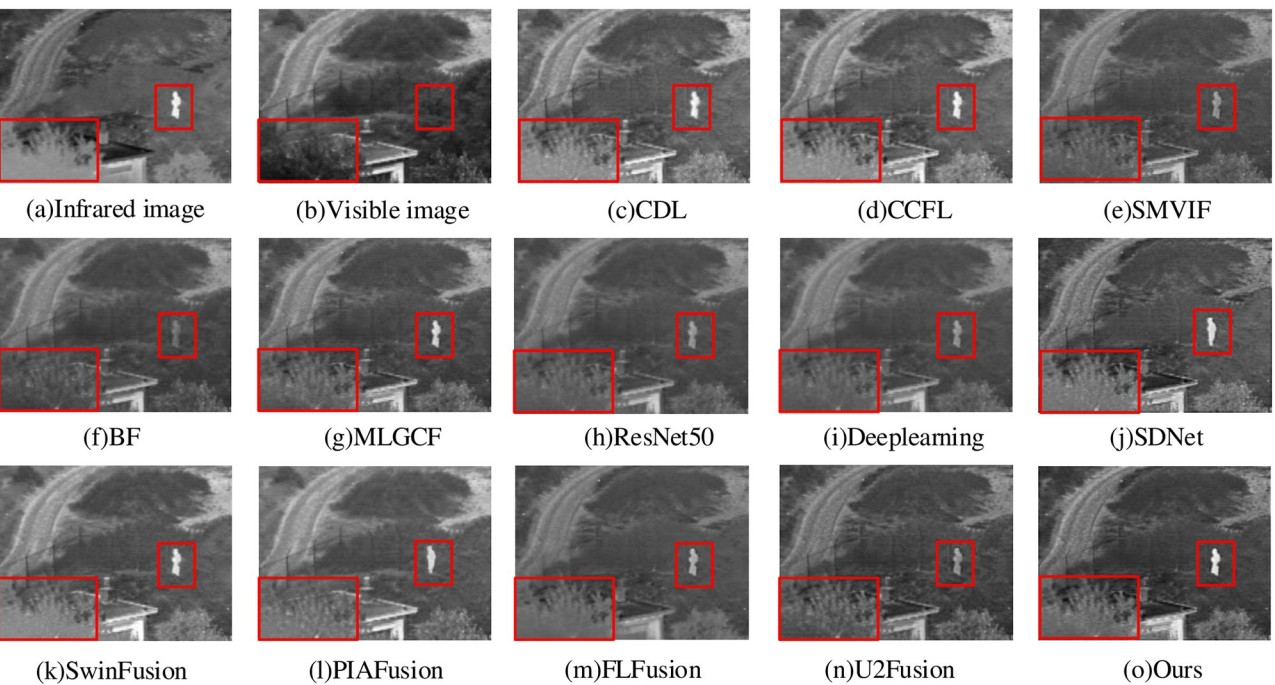

(a)Infrared image    (b)Visible image    (c)CDL    (d)CCFL    (e)SMVIF

(f)BF    (g)MLGCF    (h)ResNet50    (i)Deeplearning    (j)SDNet

(k)SwinFusion    (l)PIAFusion    (m)FLFusion    (n)U2Fusion    (o)Ours

**Fig 13. Nato camp fusion results.**

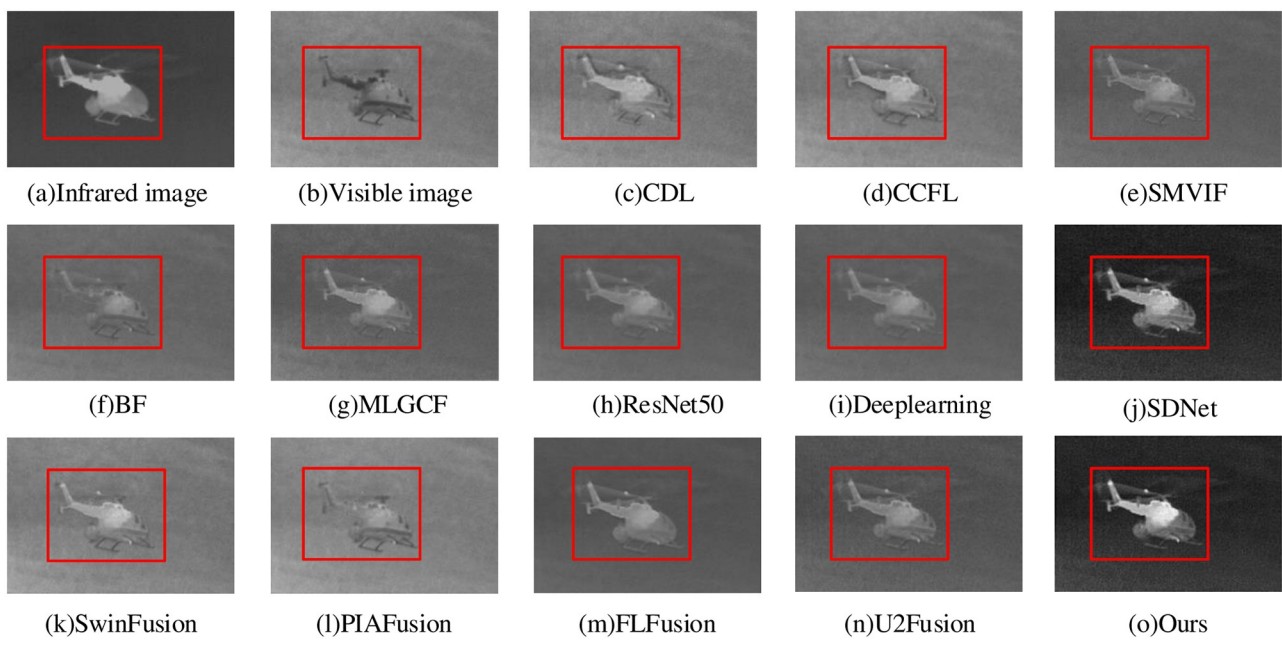

**Fig 14. Helicopter fusion results.**

The image fusion result of "Marne-04" is shown in Fig 15. The SwinFusion and PIAFusion methods exhibit a loss of cloud information from the infrared source image in the sky region. The CDL and CCFL methods contain some noise, leading to partial distortion of the image fusion result in the sky region. The SMVIF, BF, ResNet50, and Deeplearning methods do not

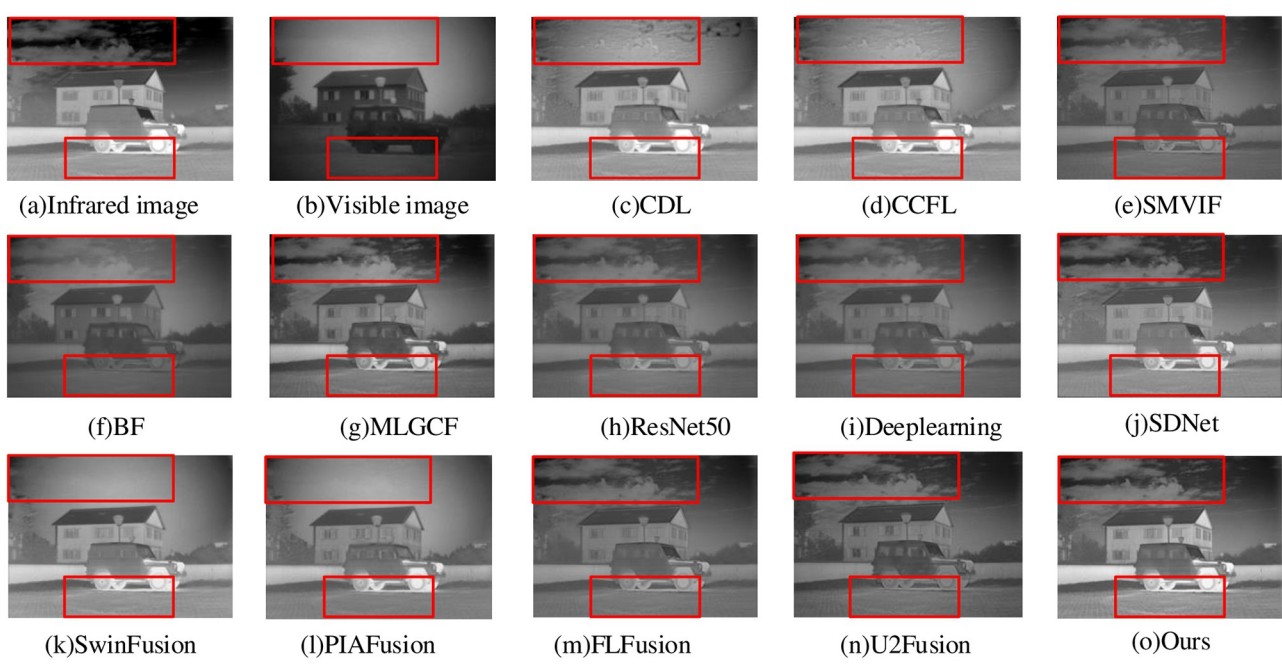

**Fig 15. Marne-04 fusion results.**

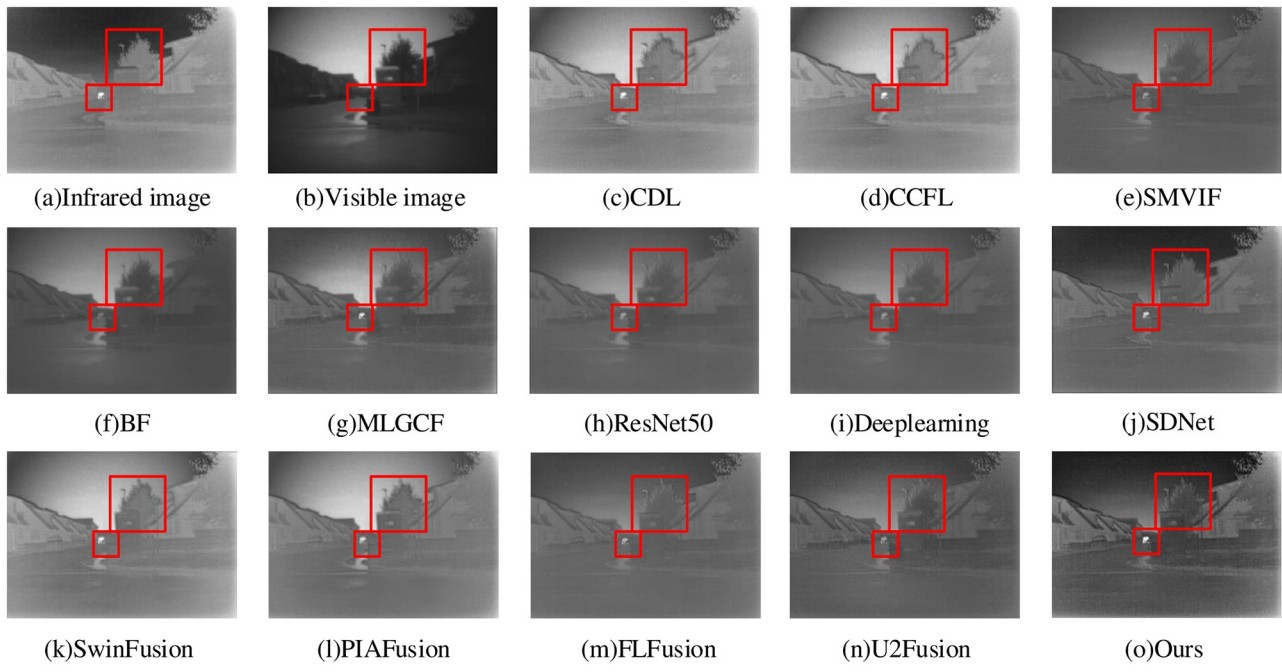

**Fig 16. Movie-01 fusion results.**

have distorted for background texture information, but the texture details on the road are lost and the overall contrast is low. The FLFusion and U2Fusion methods do not lose texture details on the road, but the window features appear blurry. The MLGCF and SDNet method have good overall contrast and meets subjective visual perception requirements, and the significant feature information (texture on the road and car windows) from the raw image is well shown in the fusion images, but more detailed regions cannot be reflected. Our proposed method could acquire good image fusion results with clearer background texture information and better brightness than previous experienced.

The image fusion result of "Movie-01" is shown in Fig 16. The overall CDL and CCFL methods provide the blurred fusion images, with the unclear trees and houses enough. The overall clarity of SwinFusion and PIAFusion method is improved, but the river in front is blurred. The visual effect of the BF method has improved, but the windows of the house are blurry and have low clarity and contrast. The SMVIF, ResNet50, Deeplearning, SDNet and FLFusion methods bring the clear object regions of the processed infrared image, with low contrast and a large amount of missing detail information. The MLGCF and U2Fusion methods have high contrast, but the object region has a virtual shadow with low clarity. Our method preserves the image feature information while having the best brightness, clarity, and detailed information.

The "Movie-18" fusion result is shown in Fig 17. The fusion image given by the BF is relatively dark with low contrast. The corresponding person and thing on the road are not clear. The ResNet50, Deeplearning, FLFusion, and U2Fusion methods have much noise and the overall fusion image is more blurred. The experimental image of the SMVIF method lacks the detailed texture information from infrared and visible images, and only contains some contour information. The CDL, CCFL, MLGCF, SwinFusion, and PIAFusion methods exhibit more prominent object regions, but there is a lack of detail information around the person. The SDNet method demonstrates good overall contrast with prominent target features, but the

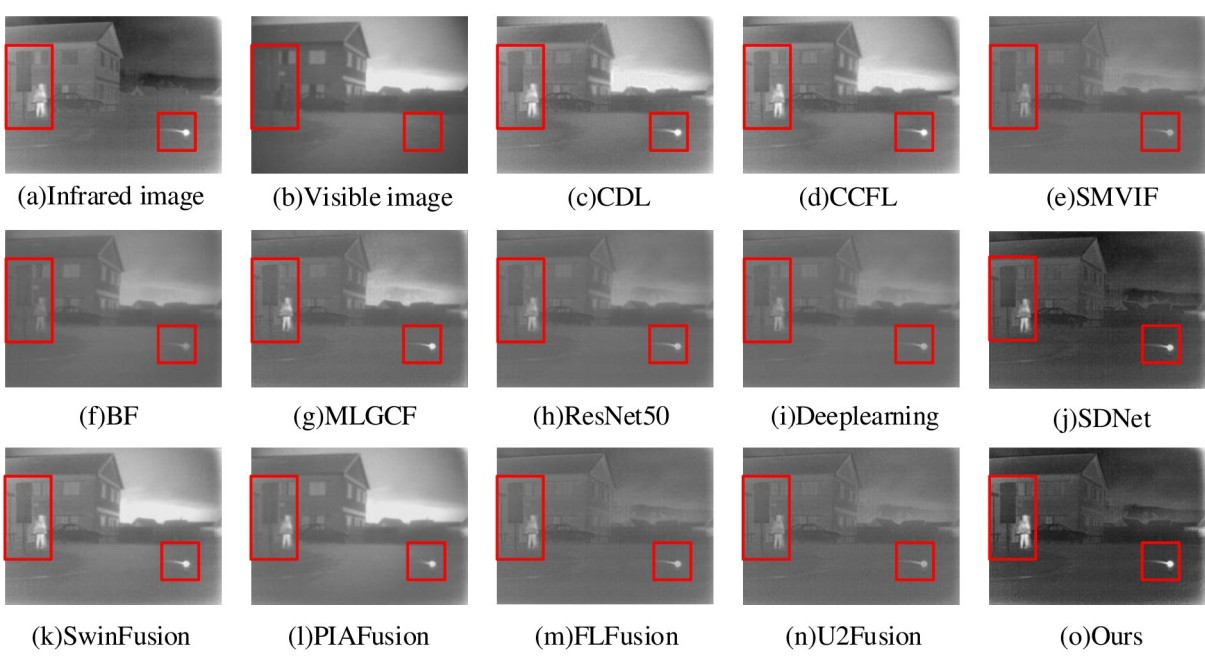

**Fig 17. Movie-18 fusion results.**

street light appears blurred. In comparison, our proposed method achieves good contrast, high definition, and detailed information, offering superior performance.

The "bench" fusion results are shown in Fig 18. The BF method has a low overall contrast with blurring flame and much noise for infrared images. The FLFusion method loses the texture detail information in the visible image, resulting in a serious lack of fusion image information. The ResNet50, the SMVIF, Deeplearning and U2Fusion have improved the overall fusion effect with rich scene information, but the flame of the object region is still not obvious with low clarity. The CDL, CCFL, SwinFusion and PIAFusion have obvious flame in the infrared images, but the clarity is not high and there is a lot of detail information loss. The SDNet and MLGCF methods achieve better overall effects, with prominent targets, but there is a slight loss of scene information. In comparison, our proposed method acquires more remarkable object regions, clearer background information, richer scene information and better visual effects.

**4.3.2 Objective evaluation.** In this paper, we select eight evaluation metrics to objectively evaluate the fused images, including average gradient (AG), entropy (EN), standard deviation (SD), spatial frequency(SF), correlation coefficient (CC), visual information fidelity for fusion (VIFF), signal to noise ratio (SNR), and mutual information (MI).

AG denotes the detailed representation and texture representation of a proceeded image by calculating the average gray-scale rate of change; EN measures the richness of the image by calculating the average information content of the image fusion result; SD reflects the separation of gray-scale values for a processed image by calculating the difference between intensity values and mean intensity values, which can helps to calculate image contrast; SF reflects the fused image sharpness by calculating the gray-level activity in the spatial domain, using information theory knowledge; MI calculates how much information the fused image includes its corresponding raw image to measure the similarity between these two images; VIFF provides the corresponding objective evaluation values of human vision system; SNR reflects the quality of

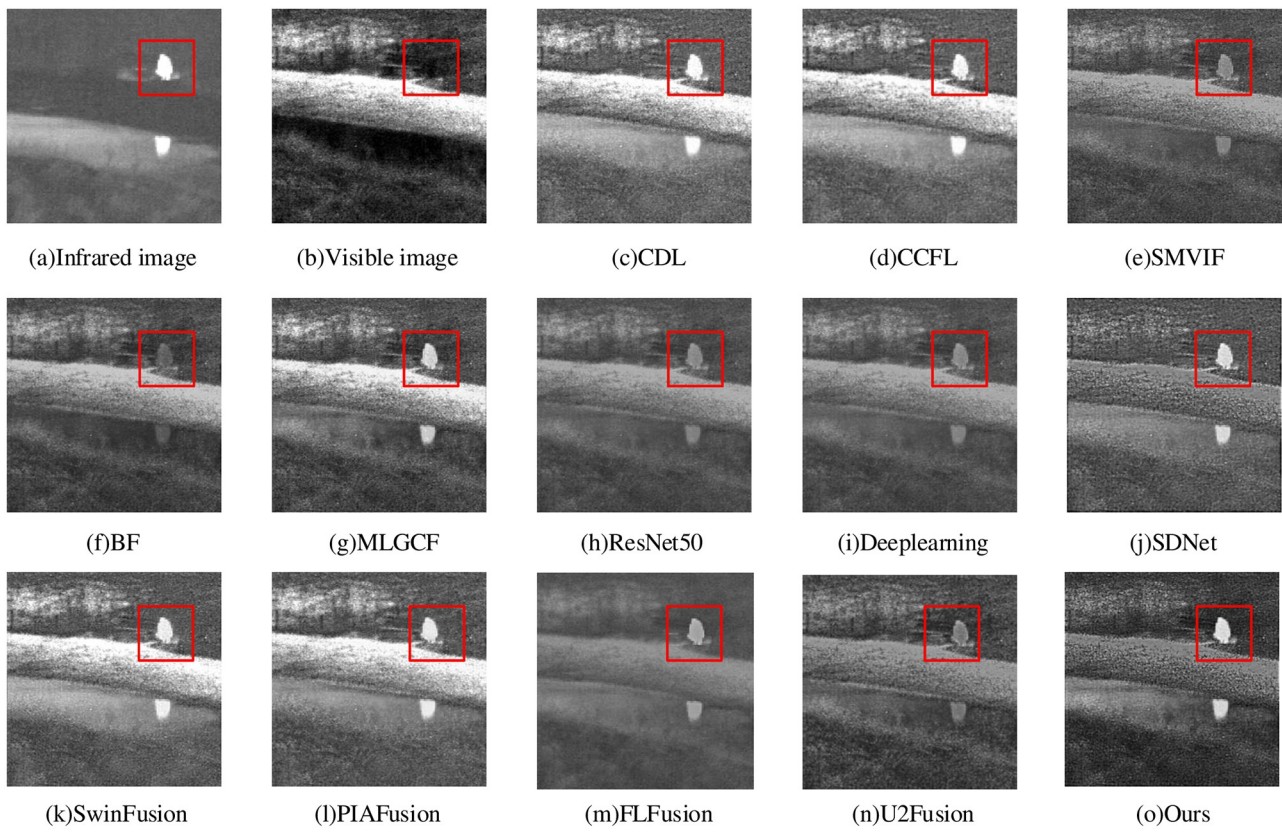

**Fig 18. Bench fusion results.**

fused images by calculating the ratio of signal to noise; CC reflects the correlation between the fused image and raw image. The evaluation results of the above metrics are shown in Tables 5–12.

Analyzing the objective evaluation metrics from Tables 5 to 12, it becomes evident that our methods exhibit high EN values, demonstrating that the image fusion result contains an

**Table 5. The quality evaluation results of the EN.**

| Method | Nato camp | helicopter | Marne-04 | Movie-01 | Movie-18 | bench | Rank |
|---|---|---|---|---|---|---|---|
| CDL | 6.7934 | 5.6257 | 7.1407 | 6.6503 | 6.7710 | 7.1787 | 12 |
| CCFL | 6.8127 | 5.6043 | 7.1047 | 6.6439 | 6.7197 | 7.1908 | 11 |
| SMVIF | 6.3308 | 5.0242 | 6.5240 | 5.5727 | 5.6541 | 6.7682 | 4 |
| BF | 6.4218 | 4.9845 | 6.4215 | 5.8274 | 5.9228 | 6.8750 | 5 |
| MLGCF | 6.6000 | 5.2277 | 7.2047 | 6.0893 | 6.0676 | 7.1909 | 7 |
| ResNet50 | 6.2505 | 4.8596 | 6.5895 | 5.5528 | 5.5639 | 6.5488 | 2 |
| Deeplearning | 6.2427 | 4.8760 | 6.4876 | 5.4801 | 5.5070 | 6.5654 | 1 |
| SDNet | 6.7067 | 5.9631 | 7.2719 | 6.2910 | 6.2731 | 6.8334 | 9 |
| SwinFusion | 6.8271 | 5.4677 | 7.0682 | 6.6539 | 6.5291 | 7.2296 | 10 |
| PIAFusion | 6.7252 | 5.3634 | 6.8790 | 6.3815 | 6.4056 | 7.2124 | 8 |
| FLFusion | 6.2736 | 4.8244 | 6.8470 | 5.7471 | 5.4241 | 6.2728 | 3 |
| U2Fusion | 6.4722 | 5.2244 | 6.7648 | 5.8975 | 5.5919 | 6.7630 | 6 |
| Ours | 6.6866 | 5.9030 | 7.5140 | 6.7814 | 6.7119 | 6.9858 | 13 |

**Table 6. The quality evaluation results of the AG.**

| Method | Nato camp | helicopter | Marne-04 | Movie-01 | Movie-18 | bench | Rank |
|---|---|---|---|---|---|---|---|
| CDL | 5.4709 | 3.4697 | 4.3026 | 3.5191 | 5.0632 | 13.4585 | 10 |
| CCFL | 4.8605 | 1.9169 | 2.5069 | 2.1355 | 2.9464 | 10.5328 | 5 |
| SMVIF | 5.2782 | 3.2055 | 3.7501 | 3.1976 | 4.5761 | 12.9586 | 9 |
| BF | 3.6227 | 2.2488 | 2.4719 | 1.6231 | 2.6759 | 9.9100 | 4 |
| MLGCF | 5.6806 | 3.3292 | 4.3630 | 3.6069 | 4.8776 | 13.4787 | 11 |
| ResNet50 | 3.2868 | 1.8002 | 2.2587 | 1.8575 | 2.6069 | 7.1408 | 2 |
| Deeplearning | 3.3059 | 1.8672 | 2.2780 | 1.8740 | 2.6004 | 7.4569 | 3 |
| SDNet | 6.4407 | 5.2537 | 4.0016 | 3.3899 | 5.8953 | 14.2100 | 12 |
| SwinFusion | 4.9732 | 2.9717 | 3.7349 | 2.9346 | 3.7870 | 13.0460 | 8 |
| PIAFusion | 4.9614 | 2.6818 | 3.1985 | 2.4141 | 3.0674 | 12.1605 | 7 |
| FLFusion | 3.4980 | 1.6990 | 2.5502 | 2.4409 | 3.2329 | 4.8610 | 1 |
| U2Fusion | 4.8742 | 2.6929 | 2.9942 | 2.9785 | 3.7734 | 9.7516 | 6 |
| Ours | 5.7896 | 5.5459 | 4.4844 | 4.3655 | 6.0008 | 14.1859 | 13 |

**Table 7. The quality evaluation results of the SD.**

| Method | Nato camp | helicopter | Marne-04 | Movie-01 | Movie-18 | bench | Rank |
|---|---|---|---|---|---|---|---|
| CDL | 5.5624 | 3.6417 | 6.1099 | 5.1180 | 5.8197 | 6.9574 | 11 |
| CCFL | 5.5693 | 3.6236 | 6.0752 | 5.1240 | 5.7747 | 6.8515 | 10 |
| SMVIF | 4.8745 | 3.1940 | 4.7778 | 3.7583 | 3.7041 | 5.6070 | 4 |
| BF | 5.1595 | 2.9414 | 4.6209 | 4.7148 | 4.5459 | 6.3362 | 6 |
| MLGCF | 5.3793 | 3.6030 | 6.0947 | 4.8090 | 4.8343 | 7.0373 | 7 |
| ResNet50 | 4.7705 | 3.1203 | 4.8803 | 3.8786 | 3.7377 | 5.4181 | 3 |
| Deeplearning | 4.7749 | 3.1163 | 4.7382 | 3.7055 | 3.6150 | 5.4068 | 1 |
| SDNet | 5.3108 | 5.2674 | 6.4815 | 5.1333 | 4.5427 | 5.4663 | 8 |
| SwinFusion | 5.7989 | 3.5937 | 5.9339 | 5.1821 | 5.6892 | 7.1252 | 12 |
| PIAFusion | 5.5968 | 3.2801 | 5.5372 | 5.1465 | 5.7962 | 7.0317 | 9 |
| FLFusion | 4.6494 | 3.4647 | 5.4063 | 3.8725 | 3.4173 | 4.6723 | 2 |
| U2Fusion | 4.9577 | 3.5121 | 5.2858 | 3.9330 | 3.5501 | 5.4379 | 5 |
| Ours | 5.5728 | 5.4033 | 6.8533 | 5.4654 | 4.8635 | 5.9775 | 13 |

**Table 8. The quality evaluation results of the SF.**

| Method | Nato camp | helicopter | Marne-04 | Movie-01 | Movie-18 | bench | Rank |
|---|---|---|---|---|---|---|---|
| CDL | 16.1523 | 8.4145 | 11.1554 | 8.8531 | 12.3207 | 34.1871 | 10 |
| CCFL | 15.2942 | 5.8435 | 8.9229 | 6.7139 | 7.8264 | 28.1212 | 6 |
| SMVIF | 15.6479 | 7.8316 | 9.5303 | 8.0119 | 10.8803 | 32.3098 | 8 |
| BF | 11.3134 | 5.3894 | 6.0003 | 4.3262 | 6.6876 | 25.2357 | 4 |
| MLGCF | 16.8741 | 8.4288 | 11.9710 | 9.7565 | 11.8888 | 34.1094 | 11 |
| ResNet50 | 9.9310 | 4.5369 | 6.1279 | 4.8429 | 6.4077 | 18.0090 | 1 |
| Deeplearning | 10.0280 | 4.6571 | 6.0775 | 4.8688 | 6.3530 | 18.9686 | 3 |
| SDNet | 19.2293 | 13.5882 | 11.3937 | 9.5623 | 14.5852 | 36.5196 | 12 |
| SwinFusion | 15.2964 | 7.4161 | 10.4395 | 8.2163 | 9.7549 | 33.2041 | 9 |
| PIAFusion | 15.3181 | 6.7163 | 9.0079 | 7.4707 | 8.1888 | 31.2299 | 7 |
| FLFusion | 10.6916 | 4.9791 | 7.8567 | 6.5462 | 7.9678 | 12.4955 | 2 |
| U2Fusion | 14.1137 | 6.9013 | 8.4888 | 7.7095 | 9.2503 | 24.4580 | 5 |
| Ours | 17.4771 | 14.4144 | 12.9037 | 11.5532 | 14.9623 | 36.2779 | 13 |

**Table 9. The quality evaluation results of the MI.**

| Method | Nato camp | helicopter | Marne-04 | Movie-01 | Movie-18 | bench | Rank |
|---|---|---|---|---|---|---|---|
| CDL | 2.1913 | 2.6782 | 3.2327 | 4.1362 | 3.0872 | 2.7215 | 11 |
| CCFL | 2.2435 | 2.4905 | 3.4634 | 4.0296 | 3.3241 | 2.3398 | 10 |
| SMVIF | 1.5235 | 1.7964 | 1.6129 | 1.6307 | 1.2814 | 1.9587 | 2 |
| BF | 2.1543 | 2.5931 | 1.9154 | 2.8839 | 2.6098 | 3.3178 | 9 |
| MLGCF | 1.7797 | 1.8656 | 2.0741 | 1.7873 | 1.6568 | 2.8348 | 5 |
| ResNet50 | 1.6226 | 2.0508 | 1.9140 | 1.8025 | 1.4937 | 2.6548 | 3 |
| Deeplearning | 1.6170 | 2.1201 | 1.8699 | 1.9183 | 1.5546 | 2.6117 | 4 |
| SDNet | 1.7916 | 1.8910 | 3.2574 | 3.1305 | 1.8633 | 1.3923 | 8 |
| SwinFusion | 2.7676 | 4.1748 | 3.7560 | 4.6161 | 3.2375 | 2.9808 | 13 |
| PIAFusion | 2.3238 | 3.8450 | 3.1340 | 4.2003 | 3.7488 | 2.8783 | 12 |
| FLFusion | 1.7752 | 2.0566 | 2.5436 | 2.3597 | 1.2733 | 2.0663 | 6 |
| U2Fusion | 1.4656 | 1.7520 | 1.7771 | 1.6224 | 1.2449 | 1.9264 | 1 |
| Ours | 1.6799 | 1.8072 | 2.8756 | 2.6705 | 1.3594 | 1.8350 | 7 |

**Table 10. The quality evaluation results of the VIFF.**

| Method | Nato camp | helicopter | Marne-04 | Movie-01 | Movie-18 | bench | Rank |
|---|---|---|---|---|---|---|---|
| CDL | 0.4261 | 0.2851 | 0.3919 | 0.2107 | 0.3401 | 0.2904 | 10 |
| CCFL | 0.4496 | 0.2064 | 0.3806 | 0.1534 | 0.2252 | 0.2853 | 8 |
| SMVIF | 0.3497 | 0.2249 | 0.2940 | 0.1384 | 0.1998 | 0.2551 | 7 |
| BF | 0.2690 | 0.1203 | 0.1505 | 0.0684 | 0.0828 | 0.2097 | 1 |
| MLGCF | 0.4375 | 0.2585 | 0.6187 | 0.2793 | 0.2952 | 0.3662 | 12 |
| ResNet50 | 0.3203 | 0.1952 | 0.2763 | 0.1198 | 0.1626 | 0.2083 | 5 |
| Deeplearning | 0.3226 | 0.1991 | 0.2720 | 0.1131 | 0.1551 | 0.2140 | 4 |
| SDNet | 0.2900 | 0.6351 | 0.4134 | 0.1556 | 0.3223 | 0.2936 | 11 |
| SwinFusion | 0.4082 | 0.2104 | 0.3467 | 0.1672 | 0.2589 | 0.2936 | 6 |
| PIAFusion | 0.2662 | 0.0764 | 0.2909 | 0.1155 | 0.1419 | 0.2723 | 3 |
| FLFusion | 0.1915 | 0.1579 | 0.3066 | 0.0844 | 0.0949 | 0.0927 | 2 |
| U2Fusion | 0.3051 | 0.2040 | 0.2474 | 0.4226 | 0.2840 | 0.3962 | 9 |
| Ours | 0.3809 | 0.6928 | 0.6900 | 0.4030 | 0.4215 | 0.3258 | 13 |

**Table 11. The quality evaluation results of the SNR.**

| Method | Nato camp | helicopter | Marne-04 | Movie-01 | Movie-18 | bench | Rank |
|---|---|---|---|---|---|---|---|
| CDL | 11.0841 | 2.1224 | 7.6876 | 8.5076 | 5.8372 | 6.0069 | 4 |
| CCFL | 11.3477 | 2.1597 | 7.5816 | 8.4969 | 5.9514 | 6.3373 | 5 |
| SMVIF | 11.5850 | 7.8623 | 8.0868 | 8.3033 | 10.6871 | 7.9034 | 7 |
| BF | 9.1953 | 5.3819 | 5.5712 | 5.7856 | 8.1658 | 5.7204 | 3 |
| MLGCF | 11.0723 | 7.5962 | 9.5227 | 7.8321 | 8.6539 | 5.6146 | 6 |
| ResNet50 | 11.6999 | 7.9143 | 8.5075 | 8.3584 | 10.6141 | 8.1164 | 9 |
| Deeplearning | 11.7553 | 7.8593 | 8.0936 | 8.3066 | 10.6867 | 8.1981 | 8 |
| SDNet | 13.7774 | 9.5390 | 15.0686 | 11.3931 | 8.8310 | 11.3330 | 12 |
| SwinFusion | 10.2281 | 2.0206 | 7.4081 | 8.3198 | 5.7563 | 5.9193 | 2 |
| PIAFusion | 8.6631 | 1.1770 | 6.6325 | 7.4046 | 5.4819 | 5.7270 | 1 |
| FLFusion | 14.3750 | 10.9385 | 10.4108 | 10.5867 | 13.2518 | 10.8263 | 13 |
| U2Fusion | 11.1309 | 10.9218 | 8.4726 | 8.2472 | 12.0021 | 7.8086 | 10 |
| Ours | 12.1429 | 10.4441 | 14.1555 | 8.9745 | 8.1024 | 7.9155 | 11 |

**Table 12. The quality evaluation results of the CC.**

| Method | Nato camp | helicopter | Marne-04 | Movie-01 | Movie-18 | bench | Rank |
|---|---|---|---|---|---|---|---|
| CDL | 0.6515 | 0.4731 | 0.6984 | 0.2804 | -0.2414 | 0.2453 | 8 |
| CCFL | 0.6683 | 0.4290 | 0.6874 | 0.2765 | -0.2379 | 0.2608 | 7 |
| SMVIF | 0.5166 | 0.7535 | 0.6949 | 0.0679 | -0.0751 | -0.0318 | 2 |
| BF | 0.2310 | 0.3447 | 0.0411 | -0.5781 | -0.5454 | -0.2765 | 1 |
| MLGCF | 0.5230 | 0.7593 | 0.7524 | 0.0136 | -0.2020 | -0.0156 | 4 |
| ResNet50 | 0.5174 | 0.7896 | 0.7546 | 0.0725 | -0.0809 | -0.0551 | 9 |
| Deeplearning | 0.5232 | 0.7812 | 0.7000 | 0.0577 | -0.1012 | -0.0420 | 3 |
| SDNet | 0.3726 | 0.9353 | 0.9699 | 0.9317 | -0.4085 | 0.4171 | 11 |
| SwinFusion | 0.5661 | 0.4734 | 0.6220 | 0.1553 | -0.3233 | 0.2191 | 5 |
| PIAFusion | 0.2771 | 0.8725 | 0.3969 | -0.2113 | -0.5134 | 0.0916 | 6 |
| FLFusion | 0.7635 | 0.9304 | 0.9095 | 0.7205 | 0.5170 | 0.3037 | 12 |
| U2Fusion | 0.5694 | 0.8446 | 0.7836 | 0.3286 | 0.2327 | 0.0192 | 10 |
| Ours | 0.7647 | 0.9407 | 0.9271 | 0.8126 | 0.7115 | 0.4531 | 13 |

abundant of information; a high SF value corresponds to high clarity in the image fusion results; the high AG value suggests that the fused image has more detailed feature and texture information; the high SD value indicates that the processed image contains abundant detailed information with high pixel intensity; the high VIFF value suggests that the subjective perception of the processed image is concordant between that of human visual system; the high SNR value suggests that the useful information in the image fusion result is retained and rarely affected by image noise; a high CC value suggests that the raw image transmit many important image features, resulting in a high correlation between the fusion result and the above features. However, our MI values are slightly lower than some comparison algorithms. This can be attributed to the fact that we employ concat and convolution fusion strategies to preserve luminance information in infrared images and texture information in visible images. The MI metric focuses mainly on the luminance information based on the mean method, if a fused image ultimately contains much noise, it will also result in the increase of luminance information. The CDL, CCFL, PIAFusion and BF methods focus on infrared information fusion while ignoring visible information, so the MI metric has the best image fusion performance.

## 5 Conclusion and future work

We propose a fusion decomposition network called FDNet to achieve the goal of image fusion. In image fusion stage, considering the large differences between raw images, a double branch fusion network framework, consisting of multi-scale layers, depthwise separable convolution and I-CBAM, is proposed on the research. Additionally, an improved Frobenius norm and adaptive gradient loss term are designed for unsupervised learning. The network framework can effectively extract image feature information while reducing computation complexity. In image decomposition stage, it is considered to decompose the fusion results to regenerate raw images, a SSIM structure loss is used as the decomposition loss. The related experimental results demonstrate that our method has high subjective visibility, good overall clarity, and clear background texture information. However, it should be noted that our image fusion framework is applicable to for aligned images, which has limitations for real-time, non-aligned images. In the future work, we will not only explore how to efficiently fuse unaligned images for real-time tasks, but also integrate to more advanced image processing techniques and design a unified fusion framework to handle other complex image fusion tasks.

## Author Contributions

**Conceptualization:** Jing Di, Jizhao Liu.

**Data curation:** Jing Di, Li Ren, Wenqing Guo, Qidong Liu.

**Formal analysis:** Jing Di.

**Methodology:** Jing Di, Li Ren, Wenqing Guo, Jing Lian.

**Software:** Jing Di, Li Ren, Huaikun Zhange.

**Writing – original draft:** Jing Di, Li Ren.

**Writing – review & editing:** Jing Di, Jing Lian.

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
