## [Decision Letter · Decision Letter 0]

22 May 2023

PONE-D-23-09775FDNet:An end-to-end fusion decomposition network for infrared and visible imagesPLOS ONE

Dear Dr. Ren,

Thank you for submitting your manuscript to PLOS ONE. After careful consideration, we feel that it has merit but does not fully meet PLOS ONE’s publication criteria as it currently stands. Therefore, we invite you to submit a revised version of the manuscript that addresses the points raised during the review process.

We look forward to receiving your revised manuscript.

Kind regards,

Chiranjibi Sitaula

Academic Editor

PLOS ONE

Journal Requirements:

"we no competing interests exist"

Additional Editor Comments:

The paper requires clarity in the presentation and novelty based on the reviewers' suggestions.

Please address them one by one.

Also, the following papers describe the CBAM attention's application in different areas, so the discussion of these works could enhance the readability of the paper as the current work is based on CBAM attention mostly.

https://link.springer.com/article/10.1007/s10489-020-02055-x

https://arxiv.org/pdf/2305.00679.pdf

Reviewers' comments:

Reviewer's Responses to Questions

**Comments to the Author**

1. Is the manuscript technically sound, and do the data support the conclusions?

Reviewer #1: Partly

Reviewer #2: Yes

2. Has the statistical analysis been performed appropriately and rigorously? 

Reviewer #1: No

Reviewer #2: Yes

3. Have the authors made all data underlying the findings in their manuscript fully available?

Reviewer #1: Yes

Reviewer #2: Yes

4. Is the manuscript presented in an intelligible fashion and written in standard English?

Reviewer #1: No

Reviewer #2: Yes

5. Review Comments to the Author

Reviewer #1: The manuscript proposes a novel infrared and visible image fusion algorithm based on fusion decomposition network. Although the experimental results show certain advantages over the compared methods, further improvement and validation are needed to make the proposed algorithm more robust and practical.

1. The writing of the manuscript needs improvement. For instance, some figures are not clear enough, some tables are not properly formatted using the standard three-line table style, and some equations are not centered.

2. The manuscript claims that the proposed fusion decomposition strategy and the adaptive fusion loss are innovative contributions to the field of infrared and visible image fusion. However, these strategies have already been proposed in SDNet.

3. Deep learning-based methods used in the comparison are not the latest or most advanced ones in the literature, and the experimental results are not convincing enough to support the superiority of the proposed algorithm. In addition, the recent SOTA methods U2Fusion, SwinFusion and SuperFusion should be discussed.

4. The authors claim that the proposed algorithm can promote high-level visual tasks. However, the manuscript lacks experimental evidence or analysis to support this claim.

5. The experimental results shown in Figures 7, 8, 9, and 10 are not easy to distinguish subjectively. Authors should consider highlighting the differences between the experimental results by zooming in on small areas of interest.

6. The authors claim in the Introduction section that all CNN-based methods for image fusion require a reference image. However, this statement is inaccurate and may mislead the readers. In the context of infrared and visible image fusion tasks, there is no reference image available. Therefore, all CNN-based IR-VIS fusion algorithms in this field are essentially unsupervised learning methods.

7. In the Abstract section, the authors claim that existing fusion algorithms have limitations in extracting multi-source information but do not specify what these limitations are. Authors should consider revising the abstract to provide more specific and concise information about the proposed algorithm and its contributions.

Reviewer #2: The work of this paper is logical. After reading this paper, I think this paper is complete and contains all the necessary components. The structure is reasonable and logical, and the design principle is clearly and moderately illustrated. To further improve this manuscript, I have the following suggestions:

1. It is recommended that the authors further refine the description of the contributions of this paper in the introduction to make it more concise and precise. By refining the description, the core contributions of this research can be more clearly conveyed, and lengthy or vague statements can be avoided.

2. The FDNet network architecture proposed in Figure 3 consists of fusion and decomposition networks. Please give a detailed explanation and justification to clarify why decomposition is needed after fusion and what is the intent of the decomposition network design.

3. Provide a brief explanation of the relationship and differences between CBAM and I_CBAM and what are the advantages of the improved I_CBAM attention mechanism.

4. Please have the authors draw a diagram to explain this paper's overall image fusion framework and the relationship between the designed gradient loss and intensity loss.

5. The designed gradient loss adaptive weight block is processed by Gaussian low-pass filtering for noise reduction, and then the optimization target of the corresponding pixel of the source image is evaluated based on the gradient richness. This process lacks visual representation, and it is recommended to insert relevant images in the manuscript to further illustrate this process for better reader comprehension.

6. I would suggest that the authors include 1-2 additional recent comparative experiments to validate the superior performance of the proposed network model. These supplementary comparative experiments can further evaluate and demonstrate the performance advantages of the proposed network model compared to other methods or models.

7. There are some errors in the manuscript's figures, equations, and other elements. Please carefully review and check them.

In summary, my recommendation for this paper is to accept it with minor revisions. Moreover, I hope the author will adjust according to the above recommendations.

6. PLOS authors have the option to publish the peer review history of their article (what does this mean?). If published, this will include your full peer review and any attached files.

Reviewer #1: No

Reviewer #2: No

---

## [Author Response · Author response to Decision Letter 0]

27 Jun 2023

To Reviewer #1:

1.The writing of the manuscript needs improvement. For instance, some figures are not clear enough, some tables are not properly formatted using the standard three-line table style, and some equations are not centered.

Thank you for your valuable comments and suggestion. According to your comments, we have used professional typesetting LaTeX software to reorganize the pictures, tables, and equations in the paper.

2. The manuscript claims that the proposed fusion decomposition strategy and the adaptive fusion loss are innovative contributions to the field of infrared and visible image fusion. However, these strategies have already been proposed in SDNet.

Thank you for your valuable comments. The overall framework of the SDNet network is divided into a fusion and a decomposition network. The fusion network performs feature extraction using dense blocks, and the decomposition network utilizes a dual-branch network to decompose the fused image, which contains rich scene information. Over the course of designing the loss functions, the intensity loss selects useful intensity information by adjusting the proportional coefficient, and the gradient loss introduces an adaptive gradient decision block. The decomposition loss utilizes the pixel-wise difference to guide the approximation between the decomposed image and the raw image. 

Our design idea inspired by the SDNet is different from the SDNet. Infrared images have a strong anti-interference ability, and visible images provide better texture detail and have high spatial resolution. Our fusion network is designed not only based on the characteristics of infrared and visible imagery, but also takes into account the different properties of raw images captured by different sensors. 

Our proposed fusion network is divided into three steps in the feature extraction process, which has an obvious difference than the SDNet acquired from dense blocks. The first step is to utilizes multi-scale convolution for shallow feature extraction. This addresses the issue of insufficient feature extraction with only a single-scale convolution kernel and enhances the network capability to extract shallow features. The second step is to use depth-separable convolution to consider the channel and spatial information in the image region separately. We employ deep convolution operations and point-by-point convolution operations to deepen the network depth without changing the size of the feature map itself. This allows for enhancing the network expression capability and constructing a lightweight network. The third step uses the improved I-CBAM attention mechanism to focus on the salient information of infrared and visible images from both the channel and spatial aspects. This process suppresses irrelevant channel information and ensures that all salient features can be effectively utilized during the fusion process. 

Our proposed decomposition network takes into account the presence of certain noise in the fused image, constructs single and double channels to extract and decompose different fusion features, and uses the constructed SSIM (structure similarity) loss function to calculate the structural similarity between the decomposition result and the raw image in terms of structural distortion, contrast distortion and luminance distortion. Hereinto, an improved Frobenius norm is chosen in the fusion loss to effectively adjust the weights between the fused image and the two raw images. This adjustment is achieved by measuring the distance between the pixel matrices of the fused image and the raw image. It prompts the network training to select more effective information. 

To demonstrate that our method is superior to the SDNet, we add the SDNet in our comparative experiments. The results can be observed in the subjective evaluation in Section 4.3.1 and the objective evaluation in Section 4.3.2. 

3. Deep learning-based methods used in the comparison are not the latest or most advanced ones in the literature, and the experimental results are not convincing enough to support the superiority of the proposed algorithm. In addition, the recent SOTA methods U2Fusion, SwinFusion and SuperFusion should be discussed.

Thank you for your valuable comments and suggestions. According to your suggestions, we have added two state-of-the-art infrared and visible image fusion algorithms, including U2Fusion and SwinFusion. However, the third SuperFusion method needs a large number of computing resources to train and tune the network. on the research, we face resource constraints and could not afford the computational cost, so we regrettably could not include the SuperFusion method in the manuscript. We sincerely apologize for this limitation and deeply understand that this is an important aspect of perfecting our research. 

In future research, we will continue to explore the SuperFusion method and will use it as a valuable direction to improve our work. In addition, we also added two latest comparison algorithms, including a progressive infrared and visible image fusion network based on illumination aware (PIAFusion) proposed by Tang et al. and a fast and lightweight infrared and visible image fusion network via feature flow and edge compensation for salient information (FLFusion) proposed by Xue et al. They can be seen in the subjective evaluation in Section 4.3.1 and the objective evaluation in Section 4.3.2. 

4. The authors claim that the proposed algorithm can promote high-level visual tasks. However, the manuscript lacks experimental evidence or analysis to support this claim.

Thank you for your valuable comments. There was an error describing our method to facilitating "high-level visual tasks" in the Abstract of the manuscript, and I would appreciate your help about pointing out this error to me. We careful checked the statement regarding "high-level visual tasks", and concluded that it should not be described as "high-level visual tasks" here, but should be described as "human eye visual characteristics". Because the subjective vision of the fused image and the labeling of the significant target regions of the fused image are primarily described in the related experiments of Section 4. This further emphasizes the high conformity of the fused image with human eye visual characteristics.

5. The experimental results shown in Figures 7, 8, 9, and 10 are not easy to distinguish subjectively. Authors should consider highlighting the differences between the experimental results by zooming in on small areas of interest.

Thank you for your valuable comments and suggestions. Based on your suggestion, we have enlarged the regions of interest in Figs. 7, 8, 9, 10 to highlight the differences between the experiments. Due to the addition of a schematic diagram illustrating the adaptive weighting process in the revised manuscript and a graphical description in the Introduction, the corresponding figure numbers have changed. Therefore, Fig. 7 becomes Fig. 9, Fig. 8 becomes Fig. 10, and so on. The revised images have been inserted into Sections 4.2.1, 4.2.2, 4.2.3, and 4.2.4 of the revised version.

6. The authors claim in the Introduction section that all CNN-based methods for image fusion require a reference image. However, this statement is inaccurate and may mislead the readers. In the context of infrared and visible image fusion tasks, there is no reference image available. Therefore, all CNN-based IR-VIS fusion algorithms in this field are essentially unsupervised learning methods.

Thank you for your valuable comments. There are indeed errors in the manuscript. Not all CNN-based image fusion algorithms are based on supervised learning. The well-known supervised learning method was only proposed in 2017 for multi-focus image fusion. In 2019, Li et al. proposed DenseFuse, which train an image reconstruction network using the COCO dataset and achieve image fusion goal.

After consulting relevant references, different from the multi-focus image fusion task, the infrared and visible image fusion task cannot generate usable labeled data for supervised training. In order to solve the problem, the network proposed later uses the CNN trained offline as a feature extractor, but lacks the ability to select or fuse deep features adaptively. This section has been revised in the revised manuscript. The revised content is given as follows:

Deep learning-based methods extract and combine image features based on strong feature learning capabilities of neural networks, and could be classified into supervised learning-based methods and unsupervised learning-based methods. Liu et al. [18] adopted the Convolutional Neural Network (CNN) for image fusion and made significant progresses comparing with traditional algorithms, but the CNN requires supervised training. For infrared and visible image fusion tasks, it is impossible to generate usable labeled data. In other words, it is impossible to artificially construct fusion images that can be referenced for supervised training. To address this problem, Li Hui et al [19] proposed to use the pre-trained VGG network for fusing infrared and visible images. This algorithm enables the extraction and fusion of multi-level deep features from the source images. Later, ResNet-50 [20] was proposed to extract and fuse depth features from the source images. However, a significant drawback of these models is their reliance on pre-trained CNN models as offline feature extractors. This limitation prevents adaptive extraction and fusion of features from the source images. Subsequently, scholars designed an end-to-end network framework specifically for image fusion. Prabhakar et al [21]. proposed an unsupervised end-to-end convolutional neural network learning framework, which does not require manually setting complex fusion strategies than other image fusion methods. The novel framework has the flexibility and versatility than previously experienced, but its performance evaluation results are not optimal for specific image fusion tasks.

7. In the Abstract section, the authors claim that existing fusion algorithms have limitations in extracting multi-source information but do not specify what these limitations are. Authors should consider revising the abstract to provide more specific and concise information about the proposed algorithm and its contributions.

Thank you for your valuable comments and suggestions. Based on the reviewer suggestions, we have added specific limitations of the existing fusion algorithms in extracting multi-source information in Abstract. The modifications are written as follows:

 However, most existing fusion methods for extracting features from infrared and visible images are based on convolutional neural networks (CNNs). These methods often fail to make full use of the salient objects and texture features in the raw image, leading to problems such as insufficient texture details and low contrast in the fused image.

To Reviewer #2:

1.It is recommended that the authors further refine the description of the contributions of this paper in the introduction to make it more concise and precise. By refining the description, the core contributions of this research can be more clearly conveyed, and lengthy or vague statements can be avoided.

Thank you for your valuable comments and suggestions. In the manuscript, the introduction section consisted of a list of each innovation point, lacking conciseness, and failing to highlight the core contributions. Based on your advice, in the revised manuscript, we have condensed and simplified the core contributions.

2.The FDNet network architecture proposed in Figure 3 consists of fusion and decomposition networks. Please give a detailed explanation and justification to clarify why decomposition is needed after fusion and what is the intent of the decomposition network design.

Thank you for your valuable comments. The proposed decomposition network in this paper aims to further improve the quality of fused images. The existing fused images have noise and redundancy to a certain extent, resulting in the lack of scene information in the fused images. The decomposition network designed in this paper constructs the structural similarity loss function SSIM by extracting and decomposing different fused feature sources, and compares the fused images with the raw images, so that the loss function is minimized and vice versa to promote the optimal training of the overall framework and finally output the final fused images.

3.Provide a brief explanation of the relationship and differences between CBAM and I-CBAM and what are the advantages of the improved I-CBAM attention mechanism.

Thank you for your valuable comments. The original Convolutional Block Attention Module (CBAM) combines spatial attention and channel attention within the convolutional module, creating a sequential attention structure from channels to spatial dimensions. However, for image fusion tasks, this sequential structure acquired channel attention feature maps based on spatial attention learning mechanism. Which partly affects the features learned by the attention module and naturally reduces the performance of the spatial attention module. 

The improved attention mechanism in this design breaks away original sequential structure, allowing the combined spatial attention module and channel attention module to learn the original input features simultaneously. This eliminates the need for the sequential order of spatial attention and channel attention, enabling direct learning of the original input feature maps. Secondly, since the performance of spatial attention is determined by the size of the convolutional kernel, the spatial attention module chooses a large 7×7 convolutional kernel to aggregate spatial features. Compared to using a smaller 3×3 convolutional kernel, this increases the size of the receptive field and the number of parameters in the module. Therefore, compared to other same receptive fields, we design a spatial attention module using dilated convolution to complete feature aggregation to reduce the number of module parameters.

4.Please have the authors draw a diagram to explain this paper's overall image fusion framework and the relationship between the designed gradient loss and intensity loss.

Thank you for your valuable comments. Based on your suggestion we have drawn a diagram to illustrate the correlation between the overall fusion framework, gradient loss and intensity loss.

5.The designed gradient loss adaptive weight block is processed by Gaussian low-pass filtering for noise reduction, and then the optimization target of the corresponding pixel of the source image is evaluated based on the gradient richness. This process lacks visual representation, and it is recommended to insert relevant images in the manuscript to further illustrate this process for better reader comprehension.

Thank you for your valuable comments and suggestions. Based on your suggestion, we have added schematic diagram of adaptive weight block corresponding to the gradient loss from Fig. 3 in Section 3.4.1.

6. I would suggest that the authors include 1-2 additional recent comparative experiments to validate the superior performance of the proposed network model. These supplementary comparative experiments can further evaluate and demonstrate the performance advantages of the proposed network model compared to other methods or models.

Thank you for your valuable comments and suggestions. Based on your suggestion we have added 2 latest deep learning comparison algorithms, including a progressive infrared and visible image fusion network based on illumination aware (PIAFusion) in Section 4.2.1 and a fast and lightweight infrared and visible image fusion network via feature flow and edge compensation for salient information (FLFusion) in Section 4.2.2.

7.There are some errors in the manuscript's figures, equations, and other elements. Please carefully review and check them.

Thank you for your valuable comments and suggestions. Based on your suggestions, we have rechecked the images, formulas, and tables in the text carefully and reformatted them.

---

## [Decision Letter · Decision Letter 1]

7 Aug 2023

FDNet:An end-to-end fusion decomposition network for infrared and visible images

PONE-D-23-09775R1

Dear Dr. Ren,

We’re pleased to inform you that your manuscript has been judged scientifically suitable for publication and will be formally accepted for publication once it meets all outstanding technical requirements.

Kind regards,

Gulistan Raja

Academic Editor

PLOS ONE

Additional Editor Comments (optional):

Reviewers' comments:

Reviewer's Responses to Questions

**Comments to the Author**

1. If the authors have adequately addressed your comments raised in a previous round of review and you feel that this manuscript is now acceptable for publication, you may indicate that here to bypass the “Comments to the Author” section, enter your conflict of interest statement in the “Confidential to Editor” section, and submit your "Accept" recommendation.

Reviewer #1: All comments have been addressed

Reviewer #2: All comments have been addressed

2. Is the manuscript technically sound, and do the data support the conclusions?

Reviewer #1: Yes

Reviewer #2: Yes

3. Has the statistical analysis been performed appropriately and rigorously? 

Reviewer #1: Yes

Reviewer #2: Yes

4. Have the authors made all data underlying the findings in their manuscript fully available?

Reviewer #1: Yes

Reviewer #2: Yes

5. Is the manuscript presented in an intelligible fashion and written in standard English?

Reviewer #1: Yes

Reviewer #2: Yes

6. Review Comments to the Author

Reviewer #1: The authors have addressed most of my concerns. In this case, I recommend accepting this paper as is.

Reviewer #2: all comments have been addressed.Once the author has revised the article to meet the publishing requirements, it is advisable to accept and publish it.

7. PLOS authors have the option to publish the peer review history of their article (what does this mean?). If published, this will include your full peer review and any attached files.

Reviewer #1: No

Reviewer #2: No

---

## [Editor Report · Acceptance letter]

7 Sep 2023

PONE-D-23-09775R1 

FDNet: An end-to-end fusion decomposition network for infrared and visible images 

Dear Dr. Ren:

I'm pleased to inform you that your manuscript has been deemed suitable for publication in PLOS ONE. Congratulations! Your manuscript is now with our production department. 

Kind regards, 

on behalf of

Dr. Gulistan Raja 

Academic Editor

PLOS ONE